# Poaceae-specific cell wall-derived oligosaccharides activate plant immunity via OsCERK1 during *Magnaporthe oryzae* infection in rice

Chao Yang[1,2,5], Rui Liu[1,2,5], Jinhuan Pang[1], Bin Ren[3], Huanbin Zhou [3], Gang Wang[4], Ertao Wang[4] & Jun Liu [1,2]✉

Many phytopathogens secrete cell wall degradation enzymes (CWDEs) to damage host cells and facilitate colonization. As the major components of the plant cell wall, cellulose and hemicellulose are the targets of CWDEs. Damaged plant cells often release damage-associated molecular patterns (DAMPs) to trigger plant immune responses. Here, we establish that the fungal pathogen *Magnaporthe oryzae* secretes the endoglucanases MoCel12A and MoCel12B during infection of rice (*Oryza sativa*). These endoglucanases target hemicellulose of the rice cell wall and release two specific oligosaccharides, namely the trisaccharide $3^1$-β-D-Cellobiosyl-glucose and the tetrasaccharide $3^1$-β-D-Cellotriosyl-glucose. $3^1$-β-D-Cellobiosyl-glucose and $3^1$-β-D-Cellotriosyl-glucose bind the immune receptor OsCERK1 but not the chitin binding protein OsCEBiP. However, they induce the dimerization of OsCERK1 and OsCEBiP. In addition, these Poaceae cell wall-specific oligosaccharides trigger a burst of reactive oxygen species (ROS) that is largely compromised in *oscerk1* and *oscebip* mutants. We conclude that $3^1$-β-D-Cellobiosyl-glucose and $3^1$-β-D-Cellotriosyl-glucose are specific DAMPs released from the hemicellulose of rice cell wall, which are perceived by an OsCERK1 and OsCEBiP immune complex during *M. oryzae* infection in rice.

[1] State Key Laboratory of Plant Genomics, Institute of Microbiology, Chinese Academy of Sciences, Beijing, China. [2] CAS Center for Excellence in Biotic Interactions, University of Chinese Academy of Sciences, Beijing, China. [3] State Key Laboratory for Biology of Plant Diseases and Insect Pests, Institute of Plant Protection, Chinese Academy of Agricultural Sciences, Beijing, China. [4] National key Laboratory of Plant Molecular Genetics, Center for Excellence in Molecular Plant Sciences, Institute of Plant Physiology and Ecology, Chinese Academy of Sciences, Shanghai, China. [5] These authors contributed equally: Chao Yang, Rui Liu. ✉email: junliu@im.ac.cn

Plants live in an environment that is exposed to diverse microorganisms, many of which are pathogenic. Phytopathogens typically carry conserved features known as pathogen-associated molecular patterns (PAMPs), including flagellin, elongation factor-Tu (EF-Tu), and peptidoglycan in bacterial pathogens, and chitin in fungal pathogens[1]. Plants have evolved an elegant immune system to recognize these non-self PAMPs and to activate immune responses as a defense against pathogen invasion. Cell surface-localized pattern recognition receptors (PRRs) perceive PAMPs and initiate PAMP-triggered immunity (PTI), which involves a burst of reactive oxygen species (ROS), $Ca^{2+}$ influx, mitogen-activated protein kinase (MAPK) activation, callose deposition, and the expression of defense-related genes[1–3].

PRRs are either plasma membrane-localized receptor-like kinases (RLKs) or receptor-like proteins[4], functioning as the first line of plant defense against pathogen infection. RLKs are typically composed of an extracellular domain, a transmembrane domain, and a cytosolic kinase domain. RLK extracellular domains are variable ectodomains featuring leucine-rich repeats (LRRs), the lysin motif (LysM), or lectin-type motifs, and are essential for PAMP recognition specificity. Two well-characterized PRRs with extracellular domains are FLAGELLIN-SENSING 2 (FLS2) and EF-Tu receptor (EFR), which perceive bacterial flagellin (or the epitope flg22) and EF-Tu (or the epitope elf18), respectively[5–7]. Different plant species may have different PAMP perception modules. In *Arabidopsis thaliana*, the LysM-RLK AtLYK5 coordinates with another LysM-RLK, CERK1, to perceive chitin and mount immune responses[8,9]. By contrast, in rice (*Oryza sativa*), OsCERK1 forms a heterodimer with OsCEBiP to perceive chitin[10].

Pathogen infection often damages host cells, resulting in the production of plant-derived damage-associated molecular patterns (DAMPs). These DAMPs function as "danger" alert signals and can activate plant immune responses[11]. The term DAMP was originally coined in studies of mammalian immunity and was subsequently extended to the studies of plant immune systems[12,13]. Currently identified plant DAMPs include proteins, carbohydrates, lipids, and nucleotides[1]. The *Arabidopsis* AtPEPs are well-established peptide DAMPs that are recognized by PEP RECEPTOR1/2, a plasma membrane-localized LRR-RLK[14,15]. PAMP-induced secreted peptides are another family of protein-derived elicitor peptides, and are suggested to be perceived by RLK7 in Arabidopsis[16]. In addition to peptide DAMPs, ATP has been found to trigger immune responses in both animal and plant cells[17,18]. Extracellular ATP (eATP) is released into the plant apoplast during cell damage and can be perceived by the PRR DORN1/LecRK-I.9, a lectin RLK[17].

The plant cell wall is comprised of a complex interconnected mixture of proteins and polysaccharides, in which cellulose microfibrils are cross-linked to the matrix polysaccharides, hemicellulose, and pectin[19,20]. Pectin-derived oligogalacturonides (OGs) are known as a major DAMP from plant cell wall[21,22]. Although cellulose and hemicellulose are the most abundant polysaccharides in the plant cell wall, the potential functions of cell wall-derived cellooligomers in plant immunity are poorly understood. Many phytopathogen genomes harbor genes that encode cell wall-degradation enzymes (CWDEs), such as polygalacturonases, cellulase, and hemicellulase, which help to break down the plant cell wall during infection[23]. Recently, the oligomers derived from cellulose are found to be perceived as DAMPs in *Arabidopsis*, but they cannot stimulate ROS burst and callose deposition; instead, they induce calcium influx and defense gene expression[24]. Similarly, the cellooligomers from fungal cell walls could also trigger plant immune responses[25,26]. Interestingly, fungal cell wall-derived cellooligomers-triggered immunity largely depends on CERK1 or operates synergistically with chitin[25,26], suggesting the essential role of CERK1 in perceiving cellooligomers.

*M. oryzae* is an economically important fungus that causes up to 30% annual losses in cereal yields. This fungus is predicted to secrete ~100 cellulose-depolymerization enzymes during infection on rice[23]. However, whether *M. oryzae* infection results in the release of DAMPs from rice cell wall is unclear. In this study, we demonstrate that *M. oryzae* secretes glycosyl hydrolases family GH12 endoglucanase MoCel12A/B to degrade rice cell walls, leading to the release of specific oligosaccharides, namely the trisaccharide $3^1$-β-D-cellobiosyl-glucose (G4G3G) and tetrasaccharide $3^1$-β-D-cellotriosyl-glucose (G4G4G3G). These specific oligosaccharides can activate immune responses in rice plants. Notably, these oligosaccharides contain a β-1,3-1,4-glucan backbone, which derives from hemicellulose and primarily presents in Poaceae species. We found that G4G3G and G4G4G3G are perceived by the plasma membrane-localized LysM-RLK OsCERK1 in rice. Thus, we reveal a set of non-cellulosic, polysaccharide-derived oligosaccharide DAMPs that can activate rice immune responses during *M. Oryzae* infection.

## Results

**MoCel12A and MoCel12B expression is induced during *M. oryzae* infection.** Ten microbial cellulolytic glycosyl hydrolase (GH) families that catalyze the degradation of plant cell wall cellulose have been characterized to date[27]. In the *M. oryzae* genome, three homologous GH12 endoglucanase-encoded genes, namely *MoCel12A*, *MoCel12B*, and *MoCel12C* corresponding to MGG_00677, MGG_08537, and MGG_10972, respectively, were identified by Takeda et al.[28] (Supplementary Fig. 1a). MoCel12A, MoCel12B, and MoCel12C are the only members of GH12 family in *M. oryzae* genome[29], where MoCel12A shares a similarity of 43.6% with MoCel12B and 32.4% with MoCel12C. *MoCel12A* transcription was significantly induced during early *M. oryzae* infection, as revealed by transcriptome assays (Supplementary Data 1)[30]. This sparked our interest because CWDEs function to break down host cell walls and are considered to proceed necrotroph growth generally[31]; however, these genes were induced during early infection (by 24 h post inoculation, hpi), a stage of biotroph growth[32,33]. To validate the transcriptome data, we examined *MoCel12A*, *MoCel12B*, and *MoCel12C* expression levels at several key infection stages. *MoCel12A* was induced by 8 hpi (Fig. 1a), the time during primary infection hyphae formed[33]. *MoCel12A* expression peaked at 24 hpi and then decreased, and was hardly detectable in the mycelium and spores (Fig. 1a). Similarly, *MoCel12B* expression was elevated during the early infection stages (Fig. 1a). By contrast, *MoCel12C* expression was relatively high in spores and the infection hyphae at 8 hpi when compared to the expression levels in the mycelium and the infection hyphae of later infection stage, suggesting that MoCel12C was also involved in infection or pathogen growth (Supplementary Fig. 1b).

**MoCel12A and MoCel12B are secreted β-glucanases.** Since MoCel12A, MoCel12B, and MoCel12C are predicted to be endoglucanases[29], we investigated their function during *M. oryzae* infection. We expressed the recombinant proteins fused to a 6 × His epitope in *Pichia pastoris* (Supplementary Fig. 1c) and then investigated the enzymatic specificity by performing enzymatic activity assays using barley β-glucan and tamarind seed xyloglucan as substrates, respectively. As expected, MoCel12A and MoCel12B hydrolyzed barley β-glucan, specifically the β-1,3-1,4-glucan polymers, with MoCel12A showing much higher activity than MoCel12B (Fig. 1b). We also expressed MoCel12A in *M. oryzae* fused to a GFP epitope, and compared their enzymatic activity with MoCel12A-His purified from *P. pastoris*. The result showed that their enzymatic activities were comparable

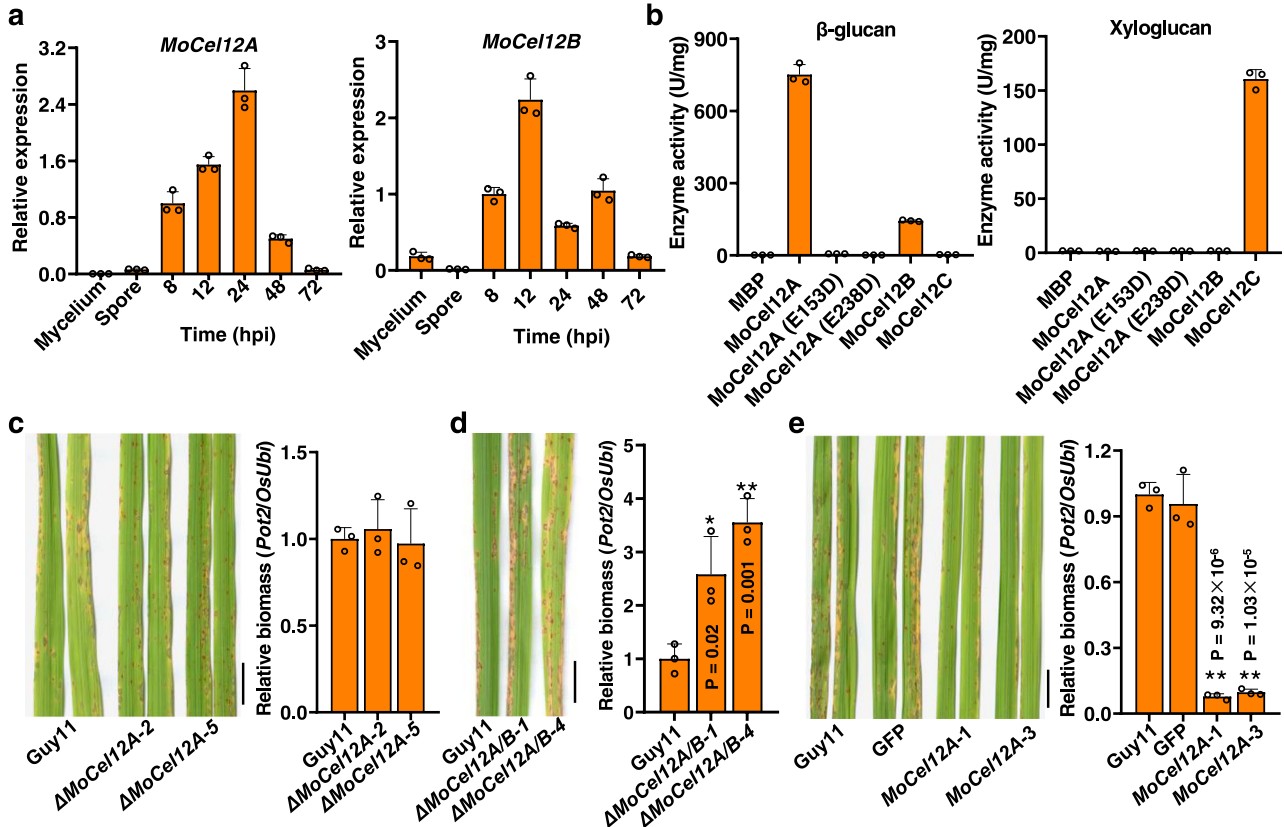

**Fig. 1 MoCel12A/B are required for the pathogenesis of *M. oryzae*. a** *MoCel12A* and *MoCel12B* transcription levels in different tissue or during *M. oryzae* infection in rice. The fungi were grown in CM media, and the mycelium and spores were collected for RT-qPCR assays. The rice leaves were inoculated with *M. oryzae* spores at a concentration of $1 \times 10^5$ per ml. The leaves were sampled at indicated time points for RT-qPCR assays. The *M. oryzae* MoActin was used as a reference gene. hpi, hours post inoculation. Values are means ± SD ($n = 3$ biological replicates). **b** The hydrolytic activity on β-glucan (left panel) and xyloglucan (right panel) of recombinant MoCel12 endoglucanases. MoCel12A (E153D) and MoCel12A (E238D) are the mutated MoCel12A. **c** Knocking out of *MoCel12A* did not impair the virulence of *M. oryzae*. Left, disease symptoms of rice leaves infected with Guy11 and two Δ*MoCel12A* mutants. Conidial suspensions ($1 \times 10^5$ conidia per ml in 0.02% Tween-20) were sprayed onto the leaf surfaces of 2-week-old rice seedlings. Images were taken at 5 dpi. Bar = 1 cm. Right, the relative fungal biomass was determined by qPCR for the *M. oryzae* Pot2 gene against rice *OsUbi* gene. Values are means ± SD ($n = 3$ biological replicates). **d** *M. oryzae* Δ*MoCel12A/B* double mutant exhibited enhanced virulence on rice. Left, disease symptoms of rice leaves infected with Guy11 and two Δ*MoCel12A/B* mutants. Right, relative fungal biomass. Others are as in (**c**). **e** Overexpression of *MoCel12A* in *M. oryzae* reduced the virulence on rice. Left, disease symptoms of rice leaves infected with Guy11, Guy11-GFP, and two independent *MoCel12A* overexpression strains. Right, relative fungal biomass. Others are as in (**c**). * and ** indicate significant differences from the treatment of Guy11 at $P < 0.05$ and 0.01, respectively (two-sided Student's *t* test).

given that MoCel12A-GFP molecular weight is twice that of MoCel12A-His (Supplementary Fig. 1d). In addition, MoCel12A did not prefer carboxymethyl cellulose (CMC), microcrystalline cellulose (MCC), or chitin as substrates (Supplementary Fig. 1e, f). By contrast, MoCel12C showed strong enzymatic activity for xyloglucan, specifically the polysaccharides of β-1,4-linked D-glucan substituted with xylose, indicating that it is a xyloglucanase. As xyloglucan is not the primary matrix polysaccharide in rice cell walls, we focused on the function of MoCel12A/B hereafter. Okada et al. reported that the key residues for *Trichoderma reesei* Cel12A endoglucanase activity are E116 and E200, which are conserved in family-12 cellulases (Supplementary Fig. 1g)[34]. We, therefore, targeted the corresponding residues in MoCel12A with the mutations E153D and E238D, and a subsequent enzymatic activity assay showed that these residue substitutions completely abolished enzyme activity, indicating that E153 and E238 are critical residues for MoCel12A function (Fig. 1b).

MoCel12A/B are putative CWDEs that are assumed to target the plant cell wall for degradation[29]. Thus, these proteins should be secreted to the host apoplast during infection. We created a version of MoCel12A with a C-terminal GFP tag that was

expressed in *M. oryzae* strain Guy11, and confirmed that it was secreted into the supernatant; however, removal of the signal peptide caused MoCel12A-GFP to remain in the pellet, suggesting that MoCel12A is a secreted protein (Supplementary Fig. 2a). In addition, we expressed MoCel12A in rice plants alongside a truncated MoCel12A version lacking the signal peptide. Both proteins were present in a crude protein extract of rice seedlings, but only MoCel12A was detected in the rice apoplastic fluids (Supplementary Fig. 2b). These results further indicate that MoCel12A is a secreted protein and that MoCel12B, by association, is also a secreted protein.

### MoCel12A/B negatively contribute to *M. oryzae* pathogenesis.
Since MoCel12A/B were shown to be induced during infection, we characterized their roles in fungal growth and development. Using a recombinant exchange technique, we created a *M. oryzae* *MoCel12A* knockout mutant (Δ*MoCel12A*; Supplementary Fig. 3a), which was analyzed alongside a *MoCel12A*-overexpressing (*MoCel12A*-OE) strain. In both of these strains, we observed normal growth and a normal cell wall stress response induced by treatment with Congo red and SDS (Supplementary Fig. 3b).

To further investigate the roles of MoCel12A and MoCel12B in *M. oryzae* pathogenesis, we created a *MoCel12A/B* double knockout mutant (Δ*MoCel12A/B*; Supplementary Fig. 3c). Analysis of cell wall stress and spore germination in the WT and Δ*MoCel12A/B* revealed comparable colony morphology and appressorium formation (Supplementary Fig. 3d, e). To examine their pathogenesis, rice plants were inoculated with Δ*MoCel12A* and Δ*MoCel12A/B* conidia. Compared to Guy11, Δ*MoCel12A* strain showed similar pathogenesis based on the resulting fungal biomass in infected plants (Fig. 1c); however, Δ*MoCel12A/B* double mutant exhibited enhanced pathogenicity compared to the WT, revealed by the more severe disease symptoms and greater fungal biomass (Fig. 1d). By contrast, *MoCel12A*-OE overexpressing strain displayed markedly impaired pathogenicity and did not efficiently cause infection in rice (Fig. 1e). Consistently, the mutant strains complemented with the native promoter-driven *MoCel12A* and *MoCel12B* restored the pathogenesis as that of wild-type strain (Supplementary Fig. 4a, b). These results imply that MoCel12A and/or MoCel12B negatively contribute to pathogen pathogenicity.

**Ecotopic expression of MoCel12A in rice enhances disease resistance**. Since MoCel12A and MoCel12B are secreted proteins, and MoCel12A exhibited considerably higher hydrolase activity than MoCel12B (Fig. 1b), we expressed MoCel12A in rice plants (Supplementary Fig. 5a) and explored whether expression of *MoCel12A* could enhance plant susceptibility to *M. oryzae* infection. Transgenic expression of *MoCel12A* in rice resulted in a dwarf phenotype (Fig. 2a). In addition, *MoCel12A* expression resulted in spontaneous lesions on the leaves of transgenic plants (Fig. 2a). By contrast, transgenic expression of *MoCel12A^{2ED}* (carrying the E153D and E238D mutations, an enzymatic activity inactive mutant) (Supplementary Fig. 5b) did not visibly affect plant growth (Fig. 2b), indicating that the phenotype was attributed to the enzymatic function of MoCel12A in Fig. 2a. Trypan-blue staining confirmed that leaf lesions were caused due to cell death (Fig. 2c), indicating a constitutive immune activation in the transgenic plants.

To verify the phenotype of *MoCel12A*-expressing plants, we examined the expression levels of several immune-responsive genes. Indeed, the expression levels of *OsRbohA*, *OsRbohD*, *OsPR3*, and *OsPR10* were significantly upregulated in *MoCel12A*-expressing plants relative to the wild-type plants, but not in *MoCel12A^{2ED}*-expressing plants (Fig. 2d). Next, we inoculated the transgenic plants with *M. oryzae* spores to assess their disease resistance. Consistently, *MoCel12A*-expressing plants displayed enhanced disease resistance compared to WT plants (Fig. 2e). By contrast, plants expressing inactive *MoCel12A^{2ED}* exhibited similar disease symptoms and fungal biomass as WT plants following infection (Fig. 2f), implying that the transgenic expression of *MoCel12A* in rice plants activates immune responses, resulting in enhanced disease resistance to *M. oryzae*.

**MoCel12A/B-hydrolyzed rice cell wall activates an immune response**. MoCel12A-activated immune response in rice could be associated with MoCel12A as a PAMP or with MoCel12A-released oligosaccharides from the rice cell wall as DAMPs. In addition, MoCel12A-released fungal cell wall components may serve as PAMPs. As *MoCel12A^{2ED}*-expressing plants did not display an autoimmune phenotype (Fig. 2b) and β-1,3-1,4-glucan barely exists in fungal cell walls, the scenario of rice cell wall-derived DAMPs seems more likely. To test this hypothesis, we first examined whether MoCel12A activated immune responses by measuring ROS bursts in rice-suspension cells. Indeed, neither the recombinant MoCel12A nor MoCel12A^{E153D}/MoCel12A^{E238D}

proteins triggered a ROS burst (Fig. 3a). Similar results were observed using the recombinant MoCel12B and MoCel12C (Supplementary Fig. 5c), indicating that MoCel12s are not PAMPs. In addition, MoCel12A-digested *M. oryzae* cell walls did not trigger immune responses in rice cells (Supplementary Fig. 5d). This result excluded the fungal cell wall-derived oligosaccharides as PAMPs. We then incubated MoCel12A with isolated rice cell walls and tested the ROS burst using the cell wall extract. The extract from MoCel12A-digested cell wall strongly activated a ROS burst, whereas extracts from inactive MoCel12A^{E153D}/MoCel12A^{E238D} digestions did not (Fig. 3b). In addition, MoCel12A digestion resulted in a stronger ROS burst than MoCel12B digestion, whereas MoCel12C digestion did not trigger a ROS burst (Fig. 3c), suggesting that MoCel12A/B hydrolyze rice cell walls, thus releasing oligosaccharides to activate the immune response.

We validated the above results by examining MAPK activation and immune-responsive gene expression. MoCel12A-digested cell wall extract induced a strong MAPK activation in rice cells, whereas MoCel12A^{E153D}/MoCel12A^{E238D} digestion did not (Fig. 3d). In addition, both MoCel12A and MoCel12B digestion induced MAPK activation, whereas MoCel12C digestion did not (Fig. 3e). Similar to MAPK activation, the expression of PTI marker genes *PBZ1* and *PAL* was markedly upregulated in rice cells exposed to MoCel12A-digested cell wall extract, but not in those digested with MoCel12A^{E153D}/MoCel12A^{E238D} (Fig. 3f). Moreover, MoCel12A- and MoCel12B-digested cell wall extracts induced *PBZ1* and *PAL* transcription, whereas MoCel12C digestion did not (Fig. 3g). These results further indicate that specific oligosaccharides released from rice cell wall by MoCel12A and MoCel12B activate plant immune responses.

**MoCel12A/B release specific oligosaccharides from rice cell wall**. We subsequently attempted to identify the oligosaccharides released by MoCel12A- and MoCel12B-hydrolyzed rice cell walls. Based on the characteristic of MoCel12A enzyme which cleaves the β-1,4 bond in the sequence of Glc-(1,3)-Glc-(1,4)-Glc (--G3G4G--), and the results from MALDI-TOF-MS analysis, we established that a trisaccharide and a tetrasaccharide are probably the two major carbohydrates released by MoCel12A- and MoCel12B-hydrolyzed rice cell wall (Fig. 4a). To further verify the oligosaccharides, we used MoCel12A to hydrolyze the highly purified barley β-glucan which is a typical component of Poaceae-specific hemicellulose and is composed of polymerized β-glucose embedded with β-1,3-1,4-glucan, and examined the molecules produced. We confirmed that MoCel12A-released oligosaccharides from barley β-glucan triggered a strong ROS burst (Supplementary Fig. 5e). As a negative control, maltose-binding protein (MBP)-incubated barley β-glucan did not trigger a ROS burst. These results suggest that the specific oligosaccharides from barley β-glucan indeed are able to activate plant immune responses. The data from MALDI-TOF-MS analysis further showed that, similar to the oligosaccharides from rice cell wall, there are two oligosaccharides, a trisaccharide, and a tetrasaccharide, responsible for the immune activation in rice (Supplementary Fig. 5f).

To characterize these specific oligosaccharides, we collected all the combinations of commercial products of trisaccharide and tetrasaccharide hydrolyzed from barley β-glucan with specific attention on the oligosaccharides containing β-1,3/1,4-glycosidic linkage. Although we hypothesized that 3^1-β-D-cellobiosyl-glucose (BGTRIB, G4G3G) and 3^1-β-D-cellotriosyl-glucose (BGTETB, G4G4G3G) were likely the two major oligosaccharides, as MoCel12A/B produce oligosaccharides with β-1,3 bonds next to the reducing end. The polymerized β-1,3-1,4-glucan could

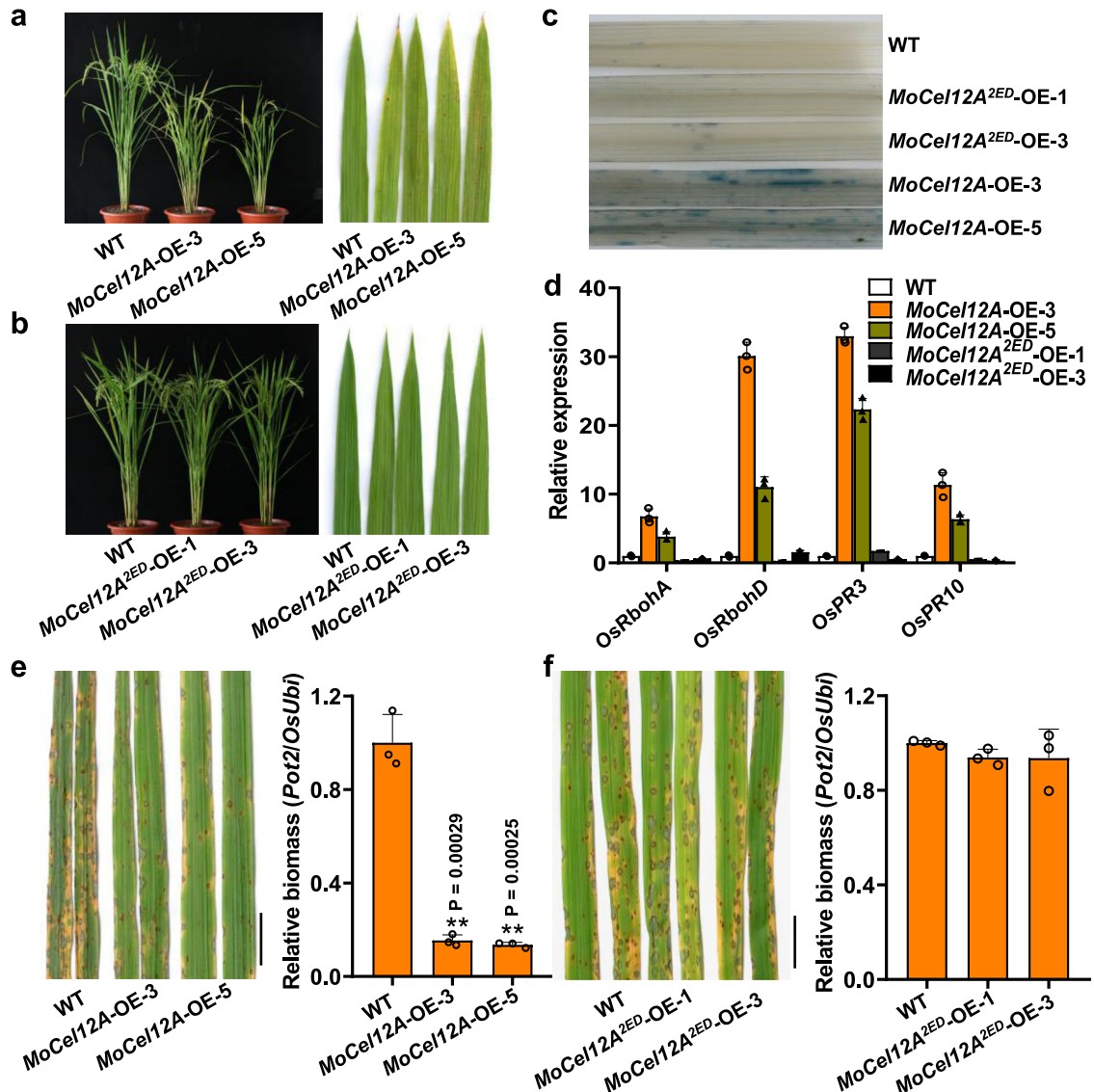

**Fig. 2 Ecotopic expression of *MoCel12A* increased the blast disease resistance in rice. a** Phenotype of the architecture (left) and the leaf lesions (right) of *MoCel12A*-expressing plants. Wild-type (WT) and *MoCel12A* ecotopic-expression plants (*MoCel12A*-OE) were grown in paddy soil and photographed at the heading stage. WT is empty vector control. OE-3 and OE-5 are two independent transgenic lines. **b** Phenotype of the architecture (Left) and the leaf lesions (Right) of *MoCel12A²ᴱᴰ* expressing plants. WT and *MoCel12A²ᴱᴰ* ecotopic-expression plants were grown as in (**a**). OE-1 and OE-3 are two independent transgenic lines. **c** Trypan-blue staining of rice plants ecotopically expressing *MoCel12A* and *MoCel12A²ᴱᴰ*. **d** *Rboh* and *PR* gene expression in WT, *MoCel12A*, and *MoCel12A²ᴱᴰ* transgenic plants. The leaves of 4-week-old plants were sampled, and RT-qPCR was used to evaluate the gene expression. Os*Actin* served as the internal reference gene. Values are means ± SD (*n* = 3 biological replicates). **e** *MoCel12A*-OE expression plants exhibit enhanced disease resistance to *M. oryzae*. Left: disease symptoms of WT and *MoCel12A*-OE leaves infected with Guy11. Conidial suspensions (1 × 10⁵ conidia per ml in 0.02% Tween-20) were sprayed onto the leaf surfaces of 2-week-old rice seedlings. The images were taken at 5 dpi. Bar = 1 cm. Right: the relative fungal biomass determined by qPCR. Values are means ± SD (*n* = 3 biological replicates). ** indicates significant differences from WT at *P* < 0.01 (two-sided Student's *t* test). **f** Expression of inactive MoCel12A in rice did not affect the blast disease resistance. Left: disease symptoms of WT and *MoCel12A²ᴱᴰ*-OE leaves infected with Guy11. Right: relative fungal biomass. Others are as in (**e**).

theoretically be hydrolyzed to several oligosaccharides by other members of the GH family. We thus examined whether 3²-β-D-glucosyl-cellobiose (BGTRIA, G3G4G) and a mixture of 3²-β-D-cellobiosyl-cellobiose and 3³-β-D-glucosyl-cellotriose (BGTETC, G4G3G4G/G3G4G4G) could activate immune responses in rice cells. In addition, we also assessed the capability of cellotriose (CTR, G4G4G) and cellotetrose (CTE, G4G4G4G), which are based on the β-1,4-glucan backbone and can be generated from cellulose, in activating immune responses. We first verified these oligosaccharides by MALDI-TOF-MS analysis (Supplementary Fig. 6a). The trisaccharides BGTRIA and BGTRIB activated

strong ROS bursts, similar to those observed in the presence of chitin, whereas CTR did not activate a ROS burst (Fig. 4b). Similarly, the tetrasaccharide BGTETB also activated a ROS burst and BGTETC activated an even stronger response than chitin, whereas CTE did not (Fig. 4c), revealing that the basic backbones of G3G4G or G4G3G are determining factors in activating ROS burst. These trisaccharide- and tetrasaccharide-triggered ROS bursts were dose-dependent (Supplementary Fig. 6b, c). In addition to the ROS assays, MAPK activation and immune-responsive gene expression assays further demonstrated that BGTRIB and BGTETB activate PTI responses in rice (Fig. 4d–f).

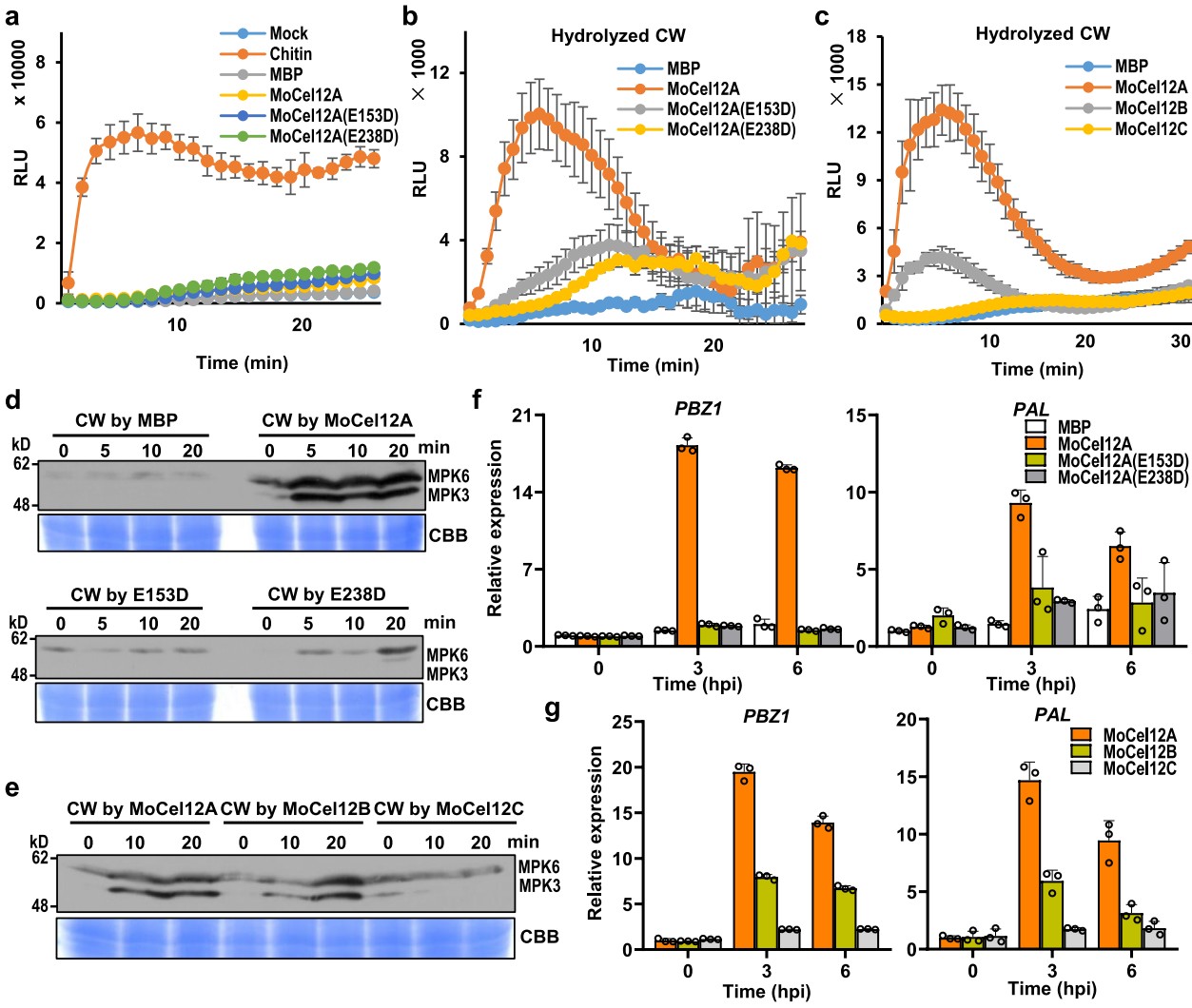

**Fig. 3 MoCel12A-hydrolyzed rice cell wall activates an immune response in rice. a** MoCel12A and it's variant proteins cannot active ROS burst in rice-suspension cells. Recombinant MoCel12A-His proteins were purified from *Pichia pastoris*, and 2 μg/ml protein each was used for the ROS burst assay. Chitin (1 μg/ml) served as a positive control. MBP served as a negative control. Values are means ± SD (*n* = 4 biological replicates). RLU relative light units. **b** MoCel12A-hydrolyzed rice cell wall activated the ROS burst in rice. Aliquots (1 ml) of a 50 mg/ml suspension of isolated rice cell walls in 50 mM sodium acetate buffer, pH 5.5, were incubated for 2 h at 37 °C with 5 μg recombinant MBP-His, MoCel12A-His, MoCel12A(E153D) -His, and MoCel12A(E238D) -His, respectively. After incubation, the supernatants (1/100, v/v) were used for the ROS burst assay. CW cell wall. Others are as in (**a**). **c** MoCel12A and MoCel12B but not MoCel12C hydrolyzed rice cell wall activated the ROS burst. The suspension of isolated rice cell walls was incubated with 5 μg recombinant MBP-His, MoCel12A-His, MoCel12B-His, and MoCel12C-His, respectively. Others are as in (**b**). **d** MoCel12A-hydrolyzed rice cell wall activated MAP kinase cascades. The supernatants of the different proteins-hydrolyzed rice cell wall were incubated with rice-suspension cells. Activated MAPKs were detected by immunoblotting with the phospho-p38 MAPK antibody at the indicated time points. The corresponding bands indicate the phosphorylation of MPK3 and MPK6. Coomassie brilliant blue (CBB) staining of Rubisco indicates equal loading of samples in each lane. The experiment was repeated three times with similar results. **e** MoCel12A and MoCel12B but not MoCel12C hydrolyzed rice cell wall activated MAP kinase cascades in rice. Others are as in (**d**). **f** MoCel12A-hydrolyzed rice cell wall induced PTI marker gene expression. Treatments with the hydrolyzed rice cell wall and rice-suspension cells were same as in (**d**), and the samples were collected for RT-qPCR at the indicated time points. Values are means ± SD (*n* = 3 biological replicates). **g** MoCel12A and MoCel12B but not MoCel12C hydrolyzed rice cell wall induced PTI marker gene expression. Others are as in (**f**).

Similar to PAMPs, pretreatment with DAMPs can prime plant disease resistance[35]. We thus tested the priming effects of BGTRIB and BGTETC on disease resistance in rice. The result showed that BGTRIB and BGTETC pretreatment enhanced plant disease resistance, exhibiting similar effects as chitin in priming defense (Fig. 4g).

We then determined the actual content of BGTRIB and BGTETB in the rice apoplastic fluid during *M. oryzae* infection. Using a high-resolution triple-quadrupole mass spectrometer, we characterized the parental and daughter M/Z as 527.3/365.1 and 689.4/347.2 for BGTRIB and BGTETB, respectively (Supplementary Fig. 7a). Based

on the parameters, we were able to identify BGTRIB and BGTETB in *M. oryzae*-infected rice apoplastic fluids (Supplementary Fig. 7b). *M. oryzae* WT infection increased BGTRIB content about twofold, while the *MoCel12A*-OE strain infection resulted in over fourfold increase in BGTRIB; however, BGTETB could be only detected in *MoCel12A*-OE strain infection, which is about twofold increase than that of Mock treatment (Supplementary Fig. 7c). These results indicate that BGTRIB and BGTETB are DAMPs released during *M. oryzae* infection. We were unable to detect BGTETC during the infection, probably because BGTETC either was not produced or the content was too low to be detected.

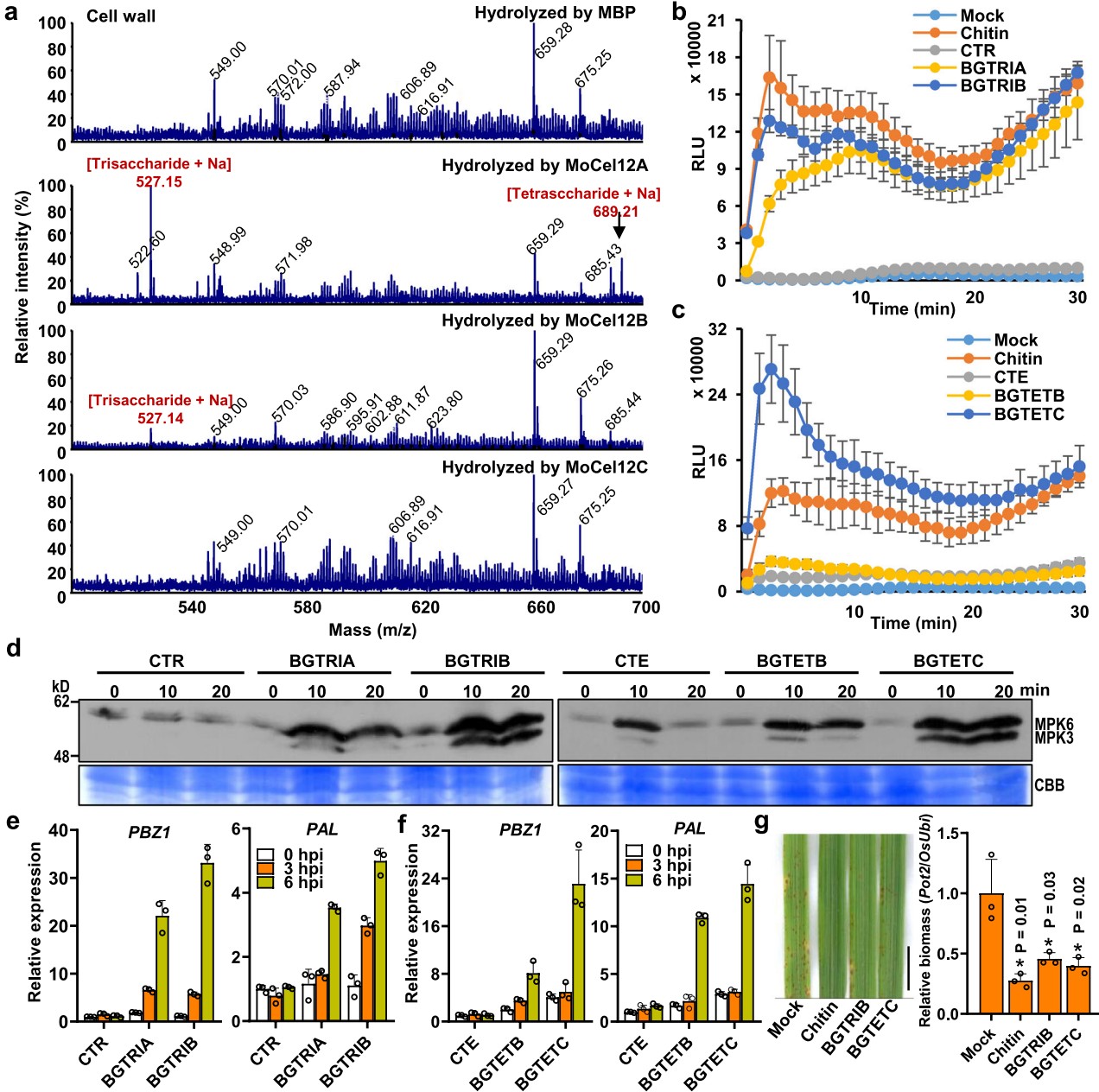

**Fig. 4 The specific trisaccharides and tetrasaccharides released by MoCel12A from rice cell wall activate the immune response in rice. a** The oligosaccharides of MoCel12A, MoCel12B, and MoCel12C hydrolyzed cell wall detected by MALDI-TOF-MS. The filtered supernatants of MoCels-hydrolyzed rice cell walls were used for the analysis, using 10 mg/ml 2,5-dihydroxybenzoic acid in 2:1 acetonitrile:water as the matrix. All indicated peaks are single-charged ions of oligosaccharides on the reducing end with Na+ using 3-aminoquinoline (3-AMQ) as ionic liquid matrix. **b** The trisaccharides activate ROS burst in rice-suspension cells. 10 μM each of CTR (cellotriose, G4G4G), BGTRIA ($3^2$-β-D-glucosyl-cellobiose, G3G4G), and BGTRIB ($3^1$-β-D-cellobiosyl-glucose, G4G3G) were used for the ROS burst assays. Chitin (10 μg/ml) served as the positive control. **c** The tetrasaccharides activate ROS burst in rice. CTE (cellotetraose, G4G4G4G), BGTETB ($3^1$-β-D-cellotriosyl-glucose, G4G4G3G), and BGTETC ($3^2$-β-D-cellobiosyl-cellobiose + $3^3$-β-D-glucosyl-cellotriose, G4G3G4G + G3G4G4G) were used for the ROS burst assays (10 μM each). Others are as in (**b**). **d** The trisaccharides and tetrasaccharides activated MAP kinase cascades in rice. The rice-suspension cells were treated with oligosaccharides (10 μM) and were used for MAPK activation assays. **e, f** The trisaccharides and tetrasaccharides induced PTI marker gene expression. Treatments with oligosaccharides and rice-suspension cells were same as in (**d**), and the samples were collected for RT-qPCR at the indicated time points. Values are means ± SD (n = 3 biological replicates). **g** Pretreatment with BGTRIB and BGTETC enhanced the blast disease resistance in rice. Two-week-old rice plants were treated with 50 μg/ml chitin, 100 μM BGTRIB and BGTETC for 3 h, then the leaves were spray-inoculated with conidial suspensions (1 × $10^5$ conidia per ml in 0.02% Tween-20) for 4 d. Bar = 1 cm. The right panel is the relative fungal biomass. Values are means ± SD (n = 3 biological replicates). * indicates significant differences from the Mock at P < 0.05 (two-sided Student's t test).

**BGTRIB- and BGTETC-triggered immune responses are compromised in *oscerk1***. Because BGTRIB and BGTETC could trigger the ROS burst in rice plants, we next sought to identify the LysM-RLK(s) which are known to perceive carbohydrates or oligosaccharides. In a screen of LysM-RLK RNAi rice-suspension cells and knockout cells, we found that BGTRIB- and BGTETC-triggered ROS bursts, as well as ROS bursts resulting from chitin, were almost undetectable in the *oscerk1* mutant (Fig. 5a). These results suggest that OsCERK1 perceives BGTRIB and BGTETC, thereby activating immune responses. In addition, we discovered that BGTRIB and BGTETC-triggered ROS bursts were partially compromised in *oscebip* mutant plants (Fig. 5b). Moreover, we found that OsRac1 was involved in the BGTRIB- and BGTETC-triggered ROS burst using an *OsRac1*-RNAi line (Supplementary Fig. 8a), which showed compromised responses to treatment with chitin and the two oligosaccharides (Fig. 5c). Notably, LYP4/6 was reported to mediate chitin recognition[36], but the mutant plants did not exhibit compromised ROS bursts following treatment with BGTRIB and BGTETC (Supplementary Fig. 8b). In addition, as OsSERK1 associates with multiple PRRs but does not seem to be involved in plant immunity[37], we used it as a negative control. The result showed that the *osserk1* mutant did not exhibit compromised ROS bursts following BGTRIB and BGTETC treatments (Supplementary Fig. 8c). These results suggest that OsCERK1 specifically perceives BGTRIB and BGTETC and that OsRac1 further transduces these signals and activates immune responses in rice.

As OsCERK1 is a typical PRR, we examined the PTI response in the *oscerk1* suspension cells treated with BGTRIB and BGTETC. The result shows that MAPK activation was markedly reduced (Supplementary Fig. 8d). Similarly, BGTRIB- and BGTETC-induced *PBZ1* and *PAL* transcription was abolished in the *oscerk1* mutant (Supplementary Fig. 8e), further indicating that OsCERK1 mediates BGTRIB and BGTETC recognition. To reinforce this conclusion, we pretreated the WT rice plants and the *oscerk1* mutant with BGTRIB and BGTETC. We found that pretreatment with the oligosaccharides increased disease resistance in WT plants but not in the *oscerk1* mutant (Supplementary Fig. 8f), implying that the *oscerk1* mutant was compromised in the perception of these specific oligosaccharides. This conclusion was further confirmed by inoculating the *oscerk1* mutant with *MoCel12A* overexpression strain, where we observed that overexpressing *MoCel12A* reduced the pathogenicity in WT plants but enhanced the pathogenicity in the mutants (Supplementary Fig. 8g).

To examine the recognition specificity of OsCERK1 for oligosaccharides, we also tested cellobiose, laminaritriose (LAM3), and xylotriose (XTR) in the activation of the ROS burst. The result showed that cellobiose and LAM3 did not trigger ROS burst in both WT and the *oscerk1* mutant; however, XTR triggered a ROS burst in WT but not *oscerk1*, indicating that OsCERK1 also recognizes XTR (Supplementary Fig. 9a). In addition, we examined whether *Arabidopsis* CERK1 functions similarly to OsCERK1. Notably, all the tested oligosaccharides except BGTETB activated a weak ROS burst and this activation was independent of CERK1 in *Arabidopsis* (Supplementary Fig. 9b). Furthermore, MoCel12A-digested *Arabidopsis* cell wall extracts could not trigger immune responses in rice (Supplementary Fig. 9c). These results demonstrate that rice cell wall-derived oligosaccharides are specific and are perceived by OsCERK1.

**BGTRIB and BGTETC bind to and activate OsCERK1**. Because OsCERK1 perceives BGTRIB and BGTETC, it may directly bind to these oligosaccharides. We, therefore, expressed the ectodomain of OsCERK1 (OsCERK1-ECD) in insect cells and purified the recombinant protein for binding assays according to the method of Liu et al.[38]. Since OsCEBiP binds to chitin and forms a heterodimer with OsCERK1, we also expressed OsCEBiP-ECD using the same method[38]. Microscale thermophoresis (MST) assays showed that OsCERK1-ECD bound to BGTETB, BGTRIB, and BGTETC (Fig. 5d). OsCERK1-ECD also bound to BGTRIA and the β-1,4-glucose-based CTR and CTE; however, it did not bind to the disaccharide sucrose and the monosaccharide glucose (Supplementary Fig. 9d). Although OsCEBiP bound to chitin, it did not bind to BGTETB, BGTRIB, and BGTETC (Fig. 5e). These results demonstrate that OsCERK1 binds to trisaccharides and tetrasaccharides, but not to disaccharides and monosaccharides.

In rice, chitin induces OsCERK1 and OsCEBiP to form a receptor complex, which transduces the chitin signal to downstream components for immune activation[10,39]. Therefore, we examined whether BGTRIB or BGTETC could induce OsCERK1 and OsCEBiP to form a receptor complex. Similar to chitin, the MoCel12A-digested rice cell extracts, BGTRIB, and BGTETC induced OsCERK1 and OsCEBiP to form a heterodimer (Fig. 5f), suggesting that these oligosaccharides share a similar immune response signaling pathway as chitin. In *Arabidopsis*, chitin binds to the LysM-RLK AtLYK5 and induces the formation of a complex with AtCERK1, thus inducing AtCERK1 phosphorylation and leading to immune activation[8]. We also observed that BGTRIB and BGTETC induced OsCERK1 homodimerization (Fig. 5g). Notably, CTR did not induce the dimerization of OsCERK1 (Supplementary Fig. 9e). These results imply that rice cell wall-derived specific oligosaccharides likely induce OsCERK1 and OsCEBiP tetramers to transduce immune signaling.

## Discussion

DAMP-activated plant immune responses are an important branch of plant innate immunity[1,40,41]. The exogenous application of many DAMPs, such as eATP or cutin monomers, triggers typical PTI responses[17,42]. So far, many DAMPs have been identified, including proteins/peptides and carbohydrates[41]. DAMPs are presumably recognized by plasma membrane-localized PRRs, but very few of the DAMP-associated PRRs have been identified[1,23,41].

Cellulose is the major component of the plant cell wall, and is formed by the polymerization of β-1,4-glucose[43]. In dicots, the non-cellulosic polysaccharides also contain a β-1,4-glucan backbone. However, in most cereal crop species, the non-cellulosic polysaccharides are heteroxylans and β-1,3-1,4-glucans[20], which are specific to grass plants. β-1,4-glucose-based cellobiose is capable of triggering calcium influx in plant cells and inducing defense gene expression[24], as well as a weak ROS burst in *Arabidopsis*[40]. However, cellotriose induced a strong defense gene expression[44]. In addition, the β-1,4 glucoside and linear β-1,3-glucan can also induce ROS burst and calcium influx in grapevine cells[45]. These lines of evidence suggest that cellobiose or cellotriose is a potent DAMP. However, it is unknown whether these oligosaccharides are generated during pathogen infection. A conserved GH17 glycosyl hydrolase from the tomato pathogen *Cladosporium fulvum* (CfGH17) was proposed to release a tomato cell wall DAMP that triggers host immune responses via an unknown plant receptor[46]. In this study, we established that the *M. oryzae*-secreted proteins MoCel12A and MoCel12B are members of the GH12 family of glycosyl hydrolases, which accept β-1,3-1,4-glucan as substrate. The resulting digestion products serve as DAMPs that activate plant immune responses during infection (Fig. 6). However, MoCel12A/B themselves are not PAMPs (Fig. 3a). Notably, the β-glucanase XEG1 produced by the soybean (*Glycine max*) pathogen *Phytophthora sojae* and the endocellulase EG1 from *Rhizoctonia solani* both exhibit cellulase activity, but they have established elicitors and their enzymatic activities are not required for elicitor activity[47,48].

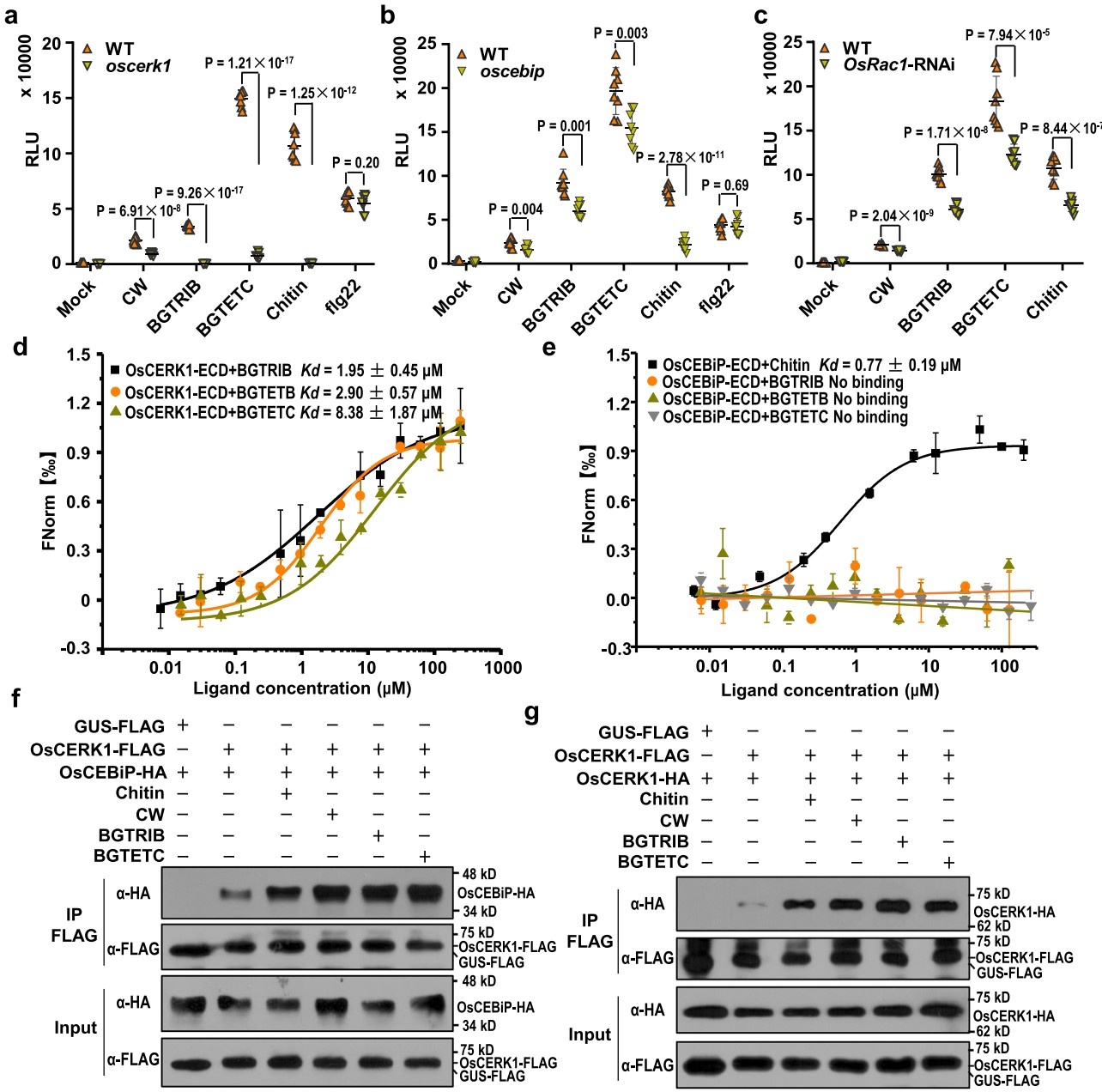

**Fig. 5 The rice cell wall-derived specific oligosaccharides are perceived by OsCERK1. a** ROS burst induced by rice cell wall-derived specific oligosaccharides is abolished in *oscerk1* mutant. The WT and *oscerk1* suspension cells were elicited with sterile water (Mock), filtered supernatants (1/100, v/v) of MoCel12A-hydrolyzed rice cell walls (CW), 10 μM BGTRIB, 10 μM BGTETC, and 5 μg/ml chitin in ROS burst assays, respectively. flg22 (100 nM) served as a positive control. Values are means ± SD (*n* = 8 biological replicates). RLU, relative light units. **b** ROS generation induced by rice cell wall-derived specific oligosaccharides in *oscebip* mutant. Others are as in (a). **c** ROS generation induced by rice cell wall-derived specific oligosaccharides in *OsRac1*-RNAi plants. Others are as in (**a**). **d** Microscale thermophoresis (MST) assays show the binding of OsCERK1-ECD with the specific oligosaccharides. The solid curve is the fit of the data points to the standard $K_d$-fit function. $K_d$, dissociation constant. Bars represent ± SD (*n* = 3 biological replicates). **e** MST assays show that OsCEBiP-ECD does not bind the oligosaccharides. Others are as in (**d**). **f** BGTRIB and BGTETC triggered heterodimerization of OsCERK1 and OsCEBiP in rice. OsCERK1-FLAG and OsCEBiP-HA were co-expressed in WT rice protoplasts. GUS-FLAG served as a negative control. The protoplasts were treated with 5 ug/ml chitin, filtered supernatants (1/100, v/v) of MoCel12a-hydrolyzed rice cell walls (CW), 10 μM BGTRIB and 10 μM BGTETC for 15 min. Co-immunoprecipitation was performed using anti-FLAG antibody and subjected to immunoblot analysis with anti-FLAG or anti-HA antibody as indicated. The experiment was repeated three times with similar results. **g** BGTRIB and BGTETC triggered OsCERK1 dimerization in rice. FLAG- and HA-tagged OsCERK1 were co-expressed in WT rice protoplasts. Others are as in (**f**).

Using a MALDI-TOF-MS and a high-resolution triple-quadrupole MS analysis approach, we identified apoplastic oligo-saccharide DAMPs from *M. oryzae*-infected rice (Supplementary Fig. 7). We established that BGTETB and BGTRIB are two major oligosaccharides released from rice cell wall during *M. oryzae* infection (Fig. 4). Enzymatic activity analysis showed that MoCel12A and MoCel12B are β-glucanases and are responsible for BGTRIB and BGTETB production during infection (Figs. 1 and 4). Furthermore, we demonstrated that hemicellulose-derived oligosaccharides BGTRIB and BGTETB strongly activate

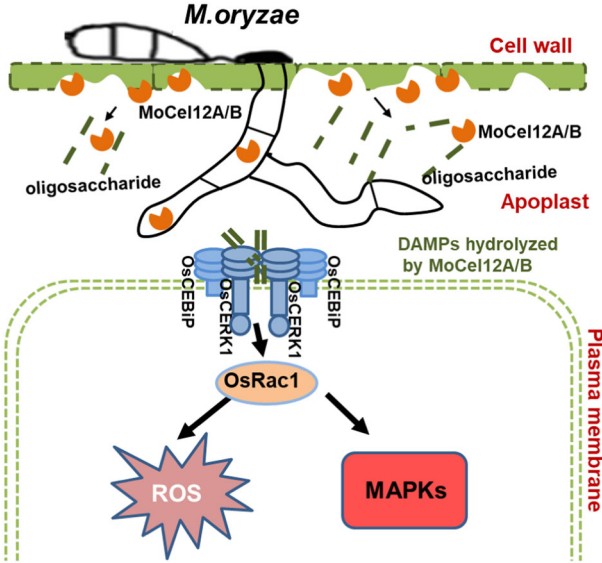

**Fig. 6 The working model.** During infection, *M. oryzae* secretes the endoglucanase MoCel12A and MoCel12B, which target rice cell wall hemicellulose for degradation. However, the oligosaccharides produced by MoCel12A and MoCel12B are perceived by immune protein complex OsCERK1-OsCEBiP, which transduces the signal to downstream OsRac1 for immune activation.

plant immune responses (Fig. 4b–f). Moreover, enhanced *M. oryzae* disease resistance results from pretreating rice plant with these oligosaccharides (Fig. 4g). Therefore, BGTETB, BGTRIB, and likely BGTETC are DAMPs that function during *M. oryzae* infection in rice.

The cell wall integrity system is essential for perceiving cell wall damage and status[41]. Souza et al. discovered that cellobiose markedly facilitated the ability of flg22 or OGs but not chitin to induce ROS burst in *Arabidopsis*[24]. Similarly, a cellotriose from the fungal cell wall induced weak ROS production in *Arabidopsis* roots; however, the combination of cellotriose and chitin induced much stronger ROS production than cellotriose or chitin alone[49], demonstrating that cellotriose functions as an amplifier for PTI. These studies reveal that PRRs are a part of the cell wall integrity system that monitors cell damage during PTI. In *Arabidopsis*, cell damage activates the expression of multiple PRRs, including FLS2, EFR, CERK1, RLP23, and LORE, which facilitate stronger immune responses during root-bacteria interactions[50]. Nevertheless, whether cellulose-derived cellooligomers are perceived by a host PRR and directly activate immune responses in plants is unknown.

Recently, the non-branched β-1,3-glucan laminarin and laminarihexaose (β-1,3-(Glc)$_6$) were shown to activate strong immune responses in two Poaceae species, namely *Hordeum vulgare* (barley) and *Brachypodium distachyon*, although the concentrations used were higher than those used here for BGTETB/C[49] (Fig. 4). Laminarin, but not laminarihexaose, also activates strong immune responses in the dicot species *Nicotiana benthamiana*, whereas laminarihexaose activates weak calcium influx and ROS production[49]. Furthermore, cell wall-derived β-1,3-D-glucans produced by the fungal pathogen *Plectosphaerella cucumerina* were recently shown to induce calcium influx and defense gene expression, and these responses are largely dependent on CERK1 in *Arabidopsis*[25]. However, laminarin-induced immune responses in the dicot *N. benthamiana* and the monocots barley and *B. distachyon* are independent of CERK1[49]. In *Arabidopsis*, CERK1

associates with LYK5 or other lysin motif PRRs such as LYK4 to perceive chitin oligosaccharides[8]. Similar to the findings by the crystal structure analysis on rice OsCERK1, *Arabidopsis* CERK1 showed a weak binding affinity to chitin[8,36], suggesting that OsCERK1 or CERK1 coordinates with chitin receptors to mediate immune activation. Using *oscerk1* mutant plants, we established that OsCERK1 is essential for the BGTETC- and BGTRIB-activated ROS burst, MAPK activation, and defense gene expression (Fig. 5a and Supplementary Fig. 8d–f). Furthermore, MST assays showed that OsCERK1 binds to BGTETB/C and BGTRIB in vitro and, importantly, BGTETC and BGTRIB can induce the homodimerization of OsCERK1-OsCERK1 and the heterodimerization of OsCERK1-OsCEBiP (Fig. 5d–g), which indicate successful recognition and immune activation[10]. Although OsCEBiP cannot bind BGTETB/C and BGTRIB, the *oscebip* mutant was partially compromised in the immune activation (Figs. 5b, e), which is likely because BGTETC and BGTRIB can induce its heterodimerization with OsCERK1 (Fig. 5f) and OsCEBiP coordinates with OsCERK1 to perceive BGTETB/C and BGTRIB. Notably, BGTRIB and BGTETB activate strong immune responses and are G4G3G-based oligosaccharides, whereas BGTRIA is G3G4G-based and triggers comparably weaker immune responses, implying that G4G3G-based oligosaccharides are the more active DAMPs. Collectively, our results suggest that β-1,4-1,3-glucose-based oligosaccharides are the potential ligands of OsCERK1. Although xyloglucan is not the substrate of MoCel12A/B (Fig. 1b), a recent study showed that the arabinoxylan-oligosaccharides can serve as DAMPs in *Arabidopsis*[51], suggesting that pathogen-secreted xyloglucanase could potentially digest plant cell wall hemicellulose to release DAMPs.

*M. oryzae* and many other fungal pathogens secrete a diverse array of CWDEs to target host cell walls for infection[23]. These CWDEs either target host cell walls for nutrition or destroy them to initiate cell death, which eventually leads to the switch of infection to necrotroph stage of this pathogen[23]. Because it is hard to distinguish the exact time course of biotroph stage, the early expression of MoCel12A/B could help for the acquisition of the nutrition or the initiation of cell death (Fig. 1a). However, the DAMPs from these events are checked by the host PRRs (Fig. 6). Because there are so many *CWDEs* in the fungal genome and many of them are functionally substitutable, removing one or two *CWDEs* generally could not significantly affect their pathogenicity[52]. In our case, overexpression of *MoCel12A* largely impaired the pathogenicity in WT plants (Fig. 1e). However, overexpression of *MoCel12A* enhanced the fungal pathogenicity in the *oscerk1* mutant, suggesting that this endoglucanase is somewhat indispensable for infection when the host cannot or is compromised to perceive the released oligosaccharides (Supplemental Fig. 8g). In fact, some PAMPs or DAMPs have been found to be modified by either host or pathogen-secreted enzymes, and the modified PAMPs or DAMPs could escape the detection by PRRs. For example, chitin oligomers and flagellin could be deacetylated and deglycosylated to evade host recognition, respectively[53,54]. In addition, the *Arabidopsis* berberine bridge enzyme-like proteins have been found to specifically oxidize the well-known DAMPs, cellulose oligomers and oligogalacturonides, and impair their elicitor activity[44,55]. Whether these mechanisms exist in the interaction between *M. oryzae* and rice to evade OsCERK1 recognition is worth further investigation.

Taken together, we have characterized an immune mechanism in which *M. oryzae* infection leads to the production of specific DAMPs in the rice apoplast, which in turn activate strong immune responses in rice plants but not in *Arabidopsis*. These species-specific DAMPs indicate the existence of a co-evolutionary plant–pathogen relationship between *M. oryzae*

and rice. Whether this specific recognition existing in other plant and pathogen interactions is worth of further investigation. In addition, how *M. oryzae* overcomes OsCERK1-mediated DAMPs recognition and the subsequent immune responses also need further investigation. Furthermore, it would be interesting to investigate why *Arabidopsis* CERK1 perceives chitin but not these specific oligosaccharides derived from rice cell walls.

## Methods

**Plant materials and growth conditions.** Rice (*Oryza sativa* subsp. Japonica cv Nipponbare) was used as the wild-type (WT) plant. The *MoCel12A*, *MoCel12A^2ED*, and *MoCel12A^nsp* (nsp, no signaling peptide) coding sequence were PCR amplified and cloned into the pCAMBIA1390 binary vector with an Ubi promoter for *Agrobacterium tumefaciens*-mediated transformation in WT plants. The *LysM-RLK* knockout or RNAi lines were described previously[56], *osserk1* from Zuo et al.[37]. The plants were grown in a growth chamber at 28 °C with a 16-h/8-h light/dark photoperiod and 75% relative humidity. The *Magnaporthe oryzae* WT strain Guy11 was cultured at 28 °C on oatmeal agar medium (oatmeal 40 g L$^{-1}$, calcium carbonate 0.6 g L$^{-1}$, agar 30 g L$^{-1}$). The conidial formation was induced under light for inoculation assays.

**Rice leaf-inoculation assays.** Rice leaf-inoculation assays were performed following the method described previously[57]. Two-week-old rice seedlings were spray-inoculated with *M. oryzae* conidial suspensions at a concentration of $1 \times 10^5$ conidia/ml in 0.02% Tween-20. Inoculated plants were placed in a growth chamber at 28 °C for 24 h in the dark, and switched back to the normal growth condition with a 16-h/8-h light/dark photoperiod. Photographs were taken at 5–7 days after inoculation. The respective fungal biomass was examined by qPCR using specific primers for *Pot2* of *M. oryzae* and normalized to the reference gene *OsUbi*. All the experiments were performed with three biological replicates.

**Gene expression analyses.** Total RNAs were isolated from rice seedlings or suspension cells using a TRIZOL reagent according to the manufacturer's instructions (Invitrogen). cDNA was synthesized using the HiScript III RT SuperMix and RT-qPCR analyses were performed on CFX96 Real-Time PCR System (BioRad) using SYBR qPCR Master Mix (Vazyme Biotech) with three biological replicates

**Fungal transformation.** *MoCel12A*, *MoCel12A^2ED*, and *MoCel12A^nsp* over-expression and *MoCel12A*, *MoCel12A* as well as *MoCel12A/B* knockout and complementary constructs, were generated as described by Li et al.[58]. Briefly, the coding sequence (CDS) amplified from *M. oryzae* were ligated with the over-expression vector pRTN-eGFP that was digested with EcoRI and BamHI. For the knockout construct, the upstream and downstream genomic sequences of *MoCel12A* or *MoCel12B* were amplified and introduced into the positions flanking the hygromycin or G418 resistance cassette of the pGKO-HPT vectors. For the complementary constructs, the native promoter (1.5 kb) and CDS of *MoCel12A* and *MoCel12B* were ligated to the complementary vectors pNat and pSUL, respectively. All the ligations were performed using a One-step Cloning Kit (Vazyme Biotech). The primers used are listed in Supplementary Table 1.

**Rice apoplastic fluid protein extraction.** The rice apoplastic fluid protein extraction was prepared according to the method described by Kim et al.[59]. Briefly, the leaves were washed with ddH$_2$O three times to remove any dust. Then, the leaves were vacuum infiltrated with ddH$_2$O under 7.5 psi for 5 min, followed by centrifugation at 1000×*g* for 5 min. The extracted apoplastic proteins were further concentrated in the Millipore tubes by the TCA-DOC method[60,61]. The proteins were re-suspended in lamelli buffer and were separated on a SDS-PAGE gel. The anti-FLAG (Sigma, F3165, 1:10,000 dilution) and anti-Rubisco (Huaxingbio, HX1989, 1:5000 dilution) antibodies were used for the immunoblotting.

**Protein expression and purification.** The *Pichia* expression system was used for the recombinant MoCel12A and its variant proteins. In brief, *pichia pastoris X-33* was selected as the host strain, and pPICZαA vector was used for secreted expression. *X-33* containing the corresponding constructs were cultured in YPD (Yeast extract–peptone–dextrose) medium. Subsequently, the culture was transferred to BMGY (buffered glycerol–complex medium) and continue to incubate for 1 d. The culture was further transferred to BMMY (buffered methanol–complex medium) for growth and induction for 5 d. Expression of the recombinant protein was induced by the addition of a final concentration of 0.5% methanol every 24 h. The fusion protein was purified using Ni-NTA Agarose (QIAGEN) according to the manufacturer's instructions.

The Bac-to-Bac baculovirus expression system was used for the recombinant OsCERK1-ECD and OsCEBiP-ECD expression. The ectodomain sequences of OsCERK1 and OsCEBiP were cloned into the baculovirus transfer vector pFastBac1 (Invitrogen) in-frame with an N-terminal gp67 signal peptide for secretion and a His6-tag at the C-terminus for purification. The recombinant

baculovirus was prepared according to the manufacturer's manual (Invitrogen). *Trichoplusia ni* (*High Five*) cells (Invitrogen) were infected with recombinant baculovirus at a multiplicity of infection of 0.5–10 at 27 °C for 48 h. The secreted soluble protein was recovered using a HisTrap HP 5-ml column (GE Healthcare).

**Enzyme activity assays.** Enzyme activities of MoCel12A and chitinase were determined following the DNS method, as previously described[62]. The barley β-glucan and tamarind seed xyloglucan (Megazyme, Ireland), as well as CMC (car-boxymethyl cellulose), MCC (microcrystalline cellulose), and chitin (Solarbio, China) were used as substrates. Chitinase from *Streptomyces griseus* was obtained from Merck (Germany). The standard reaction mixture (250 μl) containing 0.1 μg protein in 50 μl PBS buffer and 1.6 mg carbohydrates in 200 μl sodium acetate buffer (100 mM, pH 5.2). The mixture was incubated for 30 min at 37 °C, and then 500 μl DNS reagent was added. The mixture was boiled for 5 min and the color intensity was determined by the spectrophotometer at 540 nm. The chitinase activity was measured by Chitinase Assay Kit (Solarbio, China) according to the manufacturer's instructions. Briefly, 1 μg chitinase and recombinant MoCel12A were included in the reaction buffer. The enzyme activity was defined as one unit that 1 μg enzyme produces 1 μmol GlcNAc in 500 μl reaction buffer at 37 °C per hour.

**Rice cell wall hydrolysis assays.** For the rice cell wall isolation, 4-week-old rice seedlings (10–20 g) were harvested, frozen immediately in liquid nitrogen, and then were ground into powder. The material was extracted with 70% ethanol three times at 70 °C for 1 h and then dried at 50 °C to a constant weight. Aliquots (1 ml) of a 50 mg/ml suspension of the isolated rice cell wall in 50 mM sodium acetate, pH 5.5 were incubated for 2 h at 37 °C with 5 μg MoCel proteins. After incubation, samples were boiled for 20 min and centrifuged at 12,000×*g* for 30 min. The supernatant was used as the elicitors.

**Detection of the oligosaccharides in hydrolyzed cell wall.** The oligosaccharides released from MoCel12A, MoCel12B, and MoCel12C hydrolyzed cell wall were detected by MALDI-TOF-MS. MALDI-TOF-MS analysis was performed on an AB Sciex 5800 MALDI-TOF/TOF mass spectrometer. Mass spectra were obtained in the positive reflectron mode. The filtered products of different proteins-hydrolyzed rice cell walls (1 μl) were mixed with 1 μl of 2,5-dihydroxybenzoic acid solution (20 mg ml$^{-1}$ in 50:50 acetonitrile/water containing 0.1% TFA). The mixture was then spotted on the target plate with 1 μl of 5-chloro-2-mercaptobenothiazole (10 mg ml$^{-1}$) and dried for MALDI-TOF-MS analysis.

**Oxidative burst and MAPK assays.** Rice-suspension cells were used for ROS burst and MAPK assays. The cells were used 3–4 days after subcultivation in a fresh medium. The ROS burst and MAPK assays were described previously[33]. Briefly, aliquots of 500 mg cells were suspended in 5 ml pre-incubation medium (3% sucrose and 10 mM MES in 5% culture medium, pH 5.8) and incubated under culture conditions for 4–5 h. Before the assay, the medium was replaced with 200 μl fresh medium containing 20 μM L-012 (Wako Chemicals), 10 mg ml$^{-1}$ horseradish peroxidase (Sigma), and respective elicitors. The luminescence was recorded by a Centro XS3 LB 960 Luminometer (Berthold Technologies). Phospho-MPK3/6 signals were detected by a phosphorylation-specific p38 MAPK antibody (Cell Signaling, 1:5000 dilution). The highly purified oligosaccharides (CTE, CTR, BGTRIA, BGTRIB, BGTETB, BGTETC, LAM3, and XTR) used for ROS burst and MAPK activation assays were obtained from Megazyme (Ireland).

**Microscale thermophoresis (MST) analysis.** Binding reactions of recombinant OsCERK1-ECD or OsCEBiP-ECD to the oligosaccharides were measured by microscale thermophoresis in a Monolith NT.Label-Free (Nano Temper Technologies GMBH, Germany) instrument which detects changes in size, charge, and conformation induced by binding. OsCERK1-ECD or OsCEBiP-ECD (10 μM) was displaced by a buffer containing 50 mM Tris-HCl (pH 7.4), 150 mM NaCl, 10 mM MgCl$_2$ and 0.05% (v/v) Tween-20. A range of concentrations of oligosaccharides in the assay buffer (50 mM Tris-HCl pH 7.8, 150 mM NaCl, 10 mM MgCl$_2$, 0.05% Tween-20) was incubated with labeled protein (1:1, v/v) for 10 min. The sample was loaded into the NT. Label-Free standard capillaries and measured with 20% LED power and 80% MST power. The KD Fit function of the Nano Temper Analysis Software (Version 1.5.41) was used to fit the curve and calculate the value of the dissociation constant (*Kd*).

**Co-immunoprecipitation assays.** Plasmid constructs were amplified by PCR and introduced into a pUC19 vector with a HA or FLAG epitope tag at the C-terminus of OsCERK1 and OsCEBiP. Total protein was extracted from rice protoplasts transfected with OsCERK1-FLAG and OsCERK1-HA or OsCEBiP-HA using extraction/wash buffer (50 mM Tris-HCl, pH 7.5, 150 mM NaCl, 0.1% TritonX-100, 0.2% Nonidet P-40, 1 mM DTT, and 1× complete protease inhibitor (Roche)). Anti-FLAG antibody-conjugated agarose beads (Sigma) were added to the supernatant. After incubation at 4 °C on an end-over-end shaker for 1.5 h, the beads were spun down at 1500×*g* for 2 min and washed at least five times. The bound proteins were eluted by 1.5 × Laemmli loading buffer and resolved by 12%

SDS-PAGE and then subjected to immunoblot analysis using anti-FLAG antibody (Sigma, F3165, 1:10,000 dilution) or anti-HA antibody (Huxiang bio, HX1820, 1:5000 dilution).

**Phylogenetic analyses**. The amino acid sequences of the MoCel12A homologous proteins were downloaded from the UniProt website (https://www.uniprot.org/). Sequence alignment was performed with ClustalW. A neighbor-joining method implemented in MEGA7.0 was used to generate the phylogenetic tree. The bootstrap values indicated at the nodes in the phylogenetic tree are based on 1000 replications.

## Data availability
The authors declare that all data supporting the findings of this study are available within the paper and the supplementary files or are available from the corresponding author upon request. The RNA-seq data that support the findings of this study have been deposited in the SRA database of the National Center for Biotechnology Information with the accession code PRJNA707110. Mass spectrometry data that support the findings of this study have been deposited in MetaboLights with the accession code MTBLS2548. Source data are provided with this paper.

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

## Acknowledgements

We thank Ms. Yao Wu and Ms. Ying Fu at this institute for the assistance of MST and HPLC-MS/MS assays. We also thank Prof. Huishan Guo for offering us the fungi complementary vectors, and Prof. Yi Shi for the help on the protein expression using insect cells. This study was supported by the Natural Science Foundation of China (32070290, 31972257) and the Chinese Academy of Sciences Strategic Priority Research Program (Grant No. XDB11020300), the National Key R&D Program (2017YFD0200900), and a grant from the State Key Laboratory of Plant Genomics (Grant No. O8KF021011) to J.L.

## Author contributions

J.L., C.Y., and R.L. conceived and designed the experiments; C.Y. and R.L. performed most of the experiments; J.P., B.R., G.W., H.Z., and E.W. provided technical assistance; J.L. and C.Y. wrote the article.

## Competing interests

The authors declare no competing interests
