## [Peer Review File · Nature Communications]

REVIEWER COMMENTS

Reviewer #1 (Remarks to the Author):

The paper by Yang reports that oligosaccharides released from beta-1,3:1,4-gucan by endoglucanase Cel12A and B from *Magnaporthe oryzae* activate plant immunity. During the infection of bacteria and fungi, plant cells perceive small molecules from cell wall, namely damage-associated molecular patterns (DAMPs), to cause reactions for defense. Pectin oligomers (oligogalacturonides, Voxeur et al., 2019), celooligosaccharides (Souza et al., 2017), and laminarioligosaccharides have been known as DAMPs, but DMAP activity of other cell wall-derived fragments remain to be examined.

M. oryzae is a well-known fungus that causes rice blast. In this study, the authors focused on the three endoglucanases Cel12A, B and C in *M. oryzae* (MoCel12A, B and C) among ten endoglucanases. As recombinant MoCel12A and B hydrolyzed barley beta-1,3:1,4-gucan well but they hardly acted on xyloglucan, it is suggested that oligosaccharides released from beta-1,3:1,4-gucan activate immune system of rice. To test this hypothesis, the authors created MoCel12A/B knockout mutant in *M. oryzae* and observed its infection on rice. Supporting the hypothesis, the double mutant showed higher virulence than wild-type *M. oryzae*. In addition, expression of MoCel12A gene caused hypersensitive response-like lesions in transgenic plants, in which expression of pathogenesis-related genes, OsRbohA and D, OsPR3, and OsPR10, was upregulated. The authors also showed that cell wall-derived fragments possess DMAP activity but MoCel12A protein itself does not work as DAMPs. Among various glucose oligomers, beta-1,3:1,4-gucan-derived oligosaccharides such as G4G3G and G4G4G3G caused reactive-oxygen (ROS) burst as a pathogenesis-related reaction, while cellulose-based oligosaccharides, G4G4G and G4G4G4G did not. The authors also explored the perception system of oligosaccharides in rice. CERK1 and CeBiP are famous receptors that form a complex to perceive chitin oligosaccharides from fungi. It is also known that chitin oligosaccharides bind to CeBiP. Based on the facts that rice *cerk1* mutant does not respond to beta-1,3:1,4-gucan-derived oligosaccharides, the authors suggested that CERK1 is involved in the perception of the oligosaccharides. In vitro experiments using CERK1-ecto-domain (CERK1-ECD) and CEBiP expressed in insect cells, they tested the binding of oligosaccharides and interaction of these receptor proteins. Different from chitin oligosaccharides, the oligosaccharides bound to CERK1 and induced dimerization of CERK1 and CEBiP.

The present work by the authors includes several important results and contributes to understanding of plant immune system. But to improve the quality of study, the reviewer would like the authors to consider several points.

Major points

1. Release of cellobiose

It has been reported that MoCel12A and B strongly prefer beta-1,3:1,4-gucan as substrate (Takeda et al., 2010). But some of their homologues hydrolyze cellulose probably releasing cellobiose. First, the authors should examine activity of these enzymes on carboxymethyl-cellulose and crystalline cellulose. Second, release of cellobiose from barley beta-gucan should be examined using HPLC and/or TOF/MS. Third, if possible, please show the difference of responses caused by these oligosaccharides from those by cellobiose. The reviewer wonders even though cellobiose is included the hydrolysate of cell wall by MoCel12A and B, the logic and main parts of present work will not be spoiled. But if cellobiose is involved in partially, it should be mentioned.

2. Function of beta-1,3:1,4-gucan-derived oligosaccharides

It is clear that these oligosaccharides are involved in plant immune response, but the reviewer does not understand the dimerization that the authors observed is specifically caused by beta-1,3:1,4-gucan-derived oligosaccharides. As described in lines 350-351, CERK1-ECD also binds to celooligosaccharides. The authors should examine this function of other cell wall-derived fragments such as laminarioligosaccharide and xylooligosaccharide (available from Megazyme). In fact, glucuronoarabinoxylan is also a major constituent of rice cell wall as well as beta-1,3:1,4-gucan. Callose can be seen in cell wall of many plants though it exists as a minor component.

Sucrose and maltose are nice negative controls but will not show the specificity of binding.

Minor points

Beta-1,3-1,4-glucose will be beta-Glc-1,3-beta-Glc-1,4-Glc.

Properties of Cel12A and B have been studied in Taketa et al. (2010). For the priority of their work, the results on recombinant enzymes should be compared with their work.

In Fig. 4A, if possible, please include mass data for lower molecular weight as cellobiose (342+23) might be detected.

Ref.

Souza CA, Li S, Lin AZ, Boutrot F, Grossmann G, Zipfel C, Somerville SC (2017) Cellulose-Derived Oligomers Act as Damage-Associated Molecular Patterns and Trigger Defense-Like Responses. *Plant Physiol.* 173: 2383-2398.

Takeda T, Takahashi M, Nakanishi-Masuno T, Nakano Y, Saitoh H, Hirabuchi A, Fujisawa S, Terauchi R (2010) Characterization of endo-1,3-1,4- β -glucanases in GH family 12 from *Magnaporthe oryzae*. *Appl Microbiol Biotechnol.* 88: 1113-1123.

Voxeur A, Habrylo O, Guénin S, Miart F, Soulié MC, Rihouey C, Pau-Roblot C, Domon JM, Gutierrez L, Pelloux J, Mouille G, Fagard M, Höfte H, Vernhettes S (2019) Oligogalacturonide production upon *Arabidopsis thaliana*-*Botrytis cinerea* interaction. *Proc Natl Acad Sci U S A.* 116: 19743-19752.

Reviewer #2 (Remarks to the Author):

Review of Yang et al.,

During plant pathogen interactions the invading pathogen normally degrades the walls releasing oligosaccharides activating defense responses. Interest in the regulatory processes involved has been increasing steadily over the last years but our understanding of the underlying mechanisms is still limited. Here I am summarizing only the major elements of the manuscript by Yang et al.. The authors used gene expression data to identify endo-glucanases released by *Magnaporthe* during infection of rice plants. They characterized the enzymatic activities of the proteins and showed that MOLCEL12A is secreted to the apoplast. Then the authors showed that expression of active MOLCEL12A in rice increases resistance involving cell wall derived compounds whereas the inactive version did not enhance resistance. They identified cell wall derived oligosaccharides (BGTRIB, BGTETB, BGTETC) as apparently responsible for this effect, detected BGTRIB and BGTETB in the apoplast (but not BGTETC) and proceeded to show that in the *oscerk1* mutant the ROS bursts induced by BGTRIB and BGTETC are absent while they are modified in *oscebip* plants and an *OsRAC1-RNAi* line. They concluded their work by expressing the extra-cellular domains of *OsCERK1* and *OsCEBiP* to perform binding studies with the oligosaccharides and protein-protein interaction studies. The binding studies indicated that *OsCERK1* binds all three but not glucose or sucrose while *OsCEBiP* did not bind any. Intriguingly the authors also could show that binding of BGTRIB and BGTETC to *OsCERK1* caused receptor complex formation between *OsCERK1* and *OsCEBiP* as well as *OsCERK1* homodimerization.

Overall this manuscript contains a large amount of results providing insights into the mechanism underlying cell wall damage detection during infection by *Magnaporthe* in rice. The results are of general interest and elegantly connect how cell wall degradation by specific pathogen-derived enzymes induces defense responses mediated by *OsCERK1*. While I consider the results exciting and of general interest there are a few things I've listed below, which I think the authors need to address.

Major points:

Lines 184- 192: It would be great if the authors can comment / clarify the following: In line 31-32 they claimed that cell wall degrading enzymes facilitate pathogen infection. Here they show that simultaneous loss of MoCELL12A and B results in enhanced virulence fig 1 D while MoCELL12A overexpression leads to reduced virulence fig 1e? Then in figure 2C the authors show lesion phenotypes in MoCELL12A over expression lines, which seem to be different lines than the ones used in Figure 1.

Figures 3b-c and 4b-c ROS assays: I am surprised that the shape of the ROS bursts induced by cell wall extracts and the specific oligosaccharides look so different. I would assume that they should be more similar even assuming we may be looking at different concentrations of the compounds inducing them etc.

Minor points:

- The text should be proofread by an English native speaker since in some instances the wording / clarity needs to be improved. Example: line 60, lines 72-74, line 96, line 114, line 134, 163
- Line 142: I disagree with the authors. expression of MoCELL12C: expression looks elevated to me from 8-48h compared to mycelium and 72h time point.
- for completeness sake the authors should highlight in the discussion section a recent publication by Melida et al., 2020

Reviewer #3 (Remarks to the Author):

General comments:

This is an interesting work showing that two glycoside hydrolase (GH) proteins expressed by the hemibiotrophic fungal pathogen *Magnaporthe oryzae* of rice can release oligosaccharides from rice cell walls, leading to induction of defense responses. It is a very exhaustive work with a large amount of high quality data, and is beautifully written. Having said that, the MS as a whole avoids putting the proposed mechanism into biological context, and I have some fundamental questions about this. I think the individual parts of the proposed mechanism are well established, from DAMP generation to responses to perception mechanism, but I'd like to make the following points:

My main point is; if this mechanism works as proposed, then it is highly disadvantageous for *M. oryzae* to possess MoCel12A/B, and there would be strong selection against these genes. Why then are they not rapidly deleted from the *M. oryzae* genome?

The authors proposed that MoCel12A/B are required for cell wall degradation during infection. However, the double knockout is apparently able to infect just fine, which suggests to me that there is nothing wrong with its ability to degrade cell walls. I also wonder if their biomass assay is appropriate to make the claim of higher virulence, or if they should be counting spore numbers.

The authors say somewhat vaguely that this degradation activity occurs during the necrotrophic phase, but the genes are expressed early during infection when we expect the biotrophic phase. Also, DAMP perception would to my mind be active against biotrophs, but would assist necrotrophs.

The second point that I wish to make is despite the comprehensive experimentation and data in the MS, the authors have avoided investigating whether MoCel12A/B is active on fungal cell walls, for example as part of cell wall remodelling necessary for growth. I think it is essential to show that they do not degrade fungal cell walls generally, or chitin in particular. Almost all of their data can be explained by a model in which MoCel12A/B release a fungal PAMP that is perceived by CERK1 - which would be highly consistent with current models of chitin (etc) perception by this PRR. (The exception to this is the dwarf and CDR phenotypes of the MoCel12 rice OEX lines, but

the expression of the transgenes is so high in these lines that confounding effects are certainly possible).

Specific comments:

L122 "Similar DAMPs likely also exist in other Poaceae species". I am puzzled by this sentence, as the current MS doesn't seem to shed light on this. I suggest deleting it.

L127 "In the M. 128 oryzae genome, three homologous GH12 endoglucanase genes, namely MoCel12A, 129 MoCel12B, and MoCel12C corresponding to MGG_00677, MGG_08537, and 130 MGG_10972, respectively, were identified". I don't understand here what was done or what the situation is with GH genes in M. oryzae. How many GH families and genes are encoded by M. oryzae? Why were these genes in particular chosen? I can see that MoCel12A and B are closely related, but 12C does not seem to be? What is the amino acid identity between the proteins?

Fig 1/S1 What is the reference gene for RT-qPCR? It is important to know if these genes are expressed when the fungi are grown on synthetic media, please add those data (or is this what "mycelium" refers to? - if so, please make it clear).

Fig 1C and 1D need complementation lines.

L132 "...because CWDEs function to break down host cell walls, facilitating necrotrophic growth" please reference this statement. I don't understand it; necrotrophic growth occurs after the fungus has penetrated the cell wall, please comment? It is a very important point for the overall model (see my general comments).

Fig 2B Why were only two potential substrates used for these tests? I would like to see many more, particularly fungal polymers such as chitin or chitosan, and also fungal cell wall preparations. That is because the alternative hypothesis is that MoCel12A/B release a fungal PAMP. Secondly, how do the activities in Fig 2B compare to those of known B-glucanases?

L160 Would be good to mention here that the rice plants expressing Magnaporthe Mol12A are stunted.

L163 "MoCel12A/B are hydrolases that target plant cell walls for degradation" I find this statement a bit strong for this stage of the MS, please tone down (also, the double knockout seems to degrade cell walls just fine judging from Fig 1D).

L162/Fig S2B what do the Congo Red and SDS assays tell us? The MoCel12A/B double mutant should have been tested using these assays.

L188 "enhanced virulence"...I think you have to be careful with these terms. I accept that there appears to be greater fungal biomass (although I'd like to see a chitin assay too), but for greater virulence I'd like to see spore counts - this should be possible?

L221 "MoCel12A-activated rice immune responses could be associated with MoCel12A as a PAMP or with MoCel12A-hydrolyzed celooligomers from rice cell walls as DAMPs." The third hypothesis is that MoCel12A releases a fungal PAMP.

If you can knock MoCel12A/B out and enhance virulence, what stops them being lost?

Fig 4G/H- 100 uM BGTRIB/BGTETC seems very high to me, please comment? Any way to relate this to in vivo conditions under M. oryzae infection? (kind of discussed in L298-301, but concentrations are not mentioned: can the data in Fig S5C/D be converted into comparable

concentrations?).

Discussion would be improved by more of a biological discussion of how this particular mechanism interfaces with rice infection by *M. oryzae*.

Reviewer #1:

The present work by the authors includes several important results and contributes to understanding of plant immune system. But to improve the quality of study, the reviewer would like the authors to consider several points.

Response: We thank this reviewer's insightful and positive comments, and appreciate the reviewer likes this paper. We have tried our best to answer all the questions.

Comment 1. Release of cellobiose: It has been reported that MoCel12A and B strongly prefer beta-1,3,1,4-glucan as substrate (Takeda et al., 2010). But some of their homologues hydrolyze cellulose probably releasing cellobiose. First, the authors should examine activity of these enzymes on carboxymethyl-cellulose and crystalline cellulose. Second, release of cellobiose from barley beta-glucan should be examined using HPLC and/or TOF/MS. Third, if possible, please show the difference of responses caused by these oligosaccharides from those by cellobiose. The reviewer wonders even though cellobiose is included the hydrolysate of cell wall by MoCel12A and B, the logic and main parts of present work will not be spoiled. But if cellobiose is involved in partially, it should be mentioned.

Response: We have examined the enzyme activity using carboxymethyl-cellulose (CMC) and crystalline cellulose as substrate. The result showed that MoCel12A did not prefer them as substrates (Fig S1e). As the reviewer mentioned, Takeda et al. showed that MoCel12A/B prefer β -1,3,1,4-glucan as substrate, which recognize ---G3G4G---and hydrolyze the bond of G4G (2010). Therefore, most of the disaccharides (if there are any) should end with G3G but not G4G. We checked the TOF/MS data, but we did not find the peak likely to be cellobiose (below Fig 1). We also examined the disaccharides and trisaccharides in the products using HPLC-MS/MS, where we found that the disaccharides (G3G or cellobiose) are negligible compared to trisaccharides (increasing by several hundred folds, below Fig 2). Importantly, by comparing the ROS burst side by side with other oligosaccharides, the rice suspension cells did not respond to cellobiose treatment (Fig S9a). All these data suggest that cellobiose is unlikely produced by MoCel12A/B using rice cell wall as substrate.

Fig 1. Cellobiose was not detected in the MoCel12A-digested β -glucan and cell wall products.

TOF-MS data show that cellobiose was not detected at Mw 365.1. A. pure standard cellobiose; B. cellobiose was added into the control sample to exclude any possibility of fragment rearrangement. C-E. red broken line indicates the Mw 365.1 was not detected in three samples.

Fig 2. Disaccharides and trisaccharides in the MoCel12A-hydrolyzed rice cell wall.

A. The M/Z intensity of disaccharides and trisaccharides by HPLC-MS/MS in MRM mode. B and C. The separated spectrum of disaccharides and trisaccharides.

Comment 2. *Function of beta-1,3:1,4-glucan-derived oligosaccharides: It is clear that these oligosaccharides are involved in plant immune response, but the reviewer does not understand the dimerization that the authors observed is specifically caused by beta-1,3:1,4-glucan-derived oligosaccharides. As described in lines 350-351, CERK1-ECD also binds to cellooligosaccharides. The authors should examine this function of other cell wall-derived fragments such as laminarioligosaccharide and xylooligosaccharide (available from Megazyme). In fact, glucuronoarabinoxylan is also a major constituent of rice cell wall as well as beta-1,3:1,4-glucan. Callose can be seen in cell wall of many plants though it exists as a minor component. Sucrose and maltose are nice negative controls but will not show the specificity of binding.*

Response: In our case, CERK1-ECD binds β -glucan derived oligosaccharides and cellooligosaccharides (trisaccharides and tetrasaccharides) but not disaccharide and monosaccharide. These results imply CERK1-ECD selectively binds to the ligands with certain compatibility, although the crystal structure assays may give more insights. However, dimerization is crucial for immune activation of many pattern recognition receptors, which should be more specific, but binding of ligands is also required.

As suggested, we also examined laminarioligosaccharide (LAM3) and xylooligosaccharide (Xylotriose, XTR) in ROS assays. Interestingly, XTR but not LAM3 was perceived by CERK1 to trigger ROS burst (Fig S9a). The MST assays showed that both XTR and LAM3 bind CERK1-ECD although the binding affinity were lower than those specific oligosaccharides (Fig 5d and 5e, below Fig 3A). However, XTR but not cellulose and LAM3 induced the dimerization of OsCERK1 (below Fig 3B), suggesting XTR is also a potential ligand of

OsCERK1. However, we did not include the XTR story in this manuscript as we think it is not the substrate of MoCel12A and is a separated story, which may cause distraction to readers if we mentioned it in this study. As for glucuronoarabinoxylan, it is unlikely to be the substrate of endoglucanases, and unfortunately, we failed in finding the available supplier of this compound. So, we are sorry for being unable to test whether it is a substrate of MoCel12A/B. But, we hope this should not impair the merit of this study. In addition, we did not find sucrose and maltose could activate ROS burst, so we did not mention them in this study.

Fig 3: XTR binds and activates OsCERK1 dimerization.

A. Cellobiose, LAM3, and XTR bind OsCERK1 by MST assays. **B.** XTR but not cellobiose and LAM3 induced OsCERK1 dimerization.

Minor points

1. *Beta-1,3-1,4-glucose will be beta-Glc-1,3-beta-Glc-1,4-Glc.*

Response: Thanks for this comment. As suggested, we now change β -1,3-1,4-glucose to β -Glc-1,3- β -Glc-1,4-Glc when it first appears.

2. *Properties of Cel12A and B have been studied in Taketa et al. (2010). For the priority of their work, the results on recombinant enzymes should be compared with their work.*

Response: As suggested, we now included the recombinant proteins produced in *M. oryzae*, and performed the assays side by side (Fig S1d). However, we failed to generate His tagged MoCel12A in *M. oryzae*, but we generated the GFP tagged protein. The data show that the enzymatic activity of MoCel12A-His purified from yeast was about 3 times of MoCel12A-GFP from *M. oryzae*. Given the Mw of MoCel12A-GFP is about twice of MoCel12A-His, the activity of these two proteins are comparable.

3. *In Fig. 4A, if possible, please include mass data for lower molecular weight as cellobiose (342+23) might be detected.*

Response: We checked the Mass Spec data. As we respond to comment 1, we did not find any peak around 365 in all the repeats. Please refer to Fig 1 in this letter.

Reviewer #2:

During plant pathogen interactions the invading pathogen normally degrades the walls releasing oligosaccharides activating defense responses. Interest in the regulatory processes involved has been increasing steadily over the last years but our understanding of the

underlying mechanisms is still limited. Overall this manuscript contains a large amount of results providing insights into the mechanism underlying cell wall damage detection during infection by Magnaporthe in rice. The results are of general interest and elegantly connect how cell wall degradation by specific pathogen-derived enzymes induces defense responses mediated by OsCERK1. While I consider the results exciting and of general interest there are a few things I've listed below, which I think the authors need to address.

Response: Thanks for the positive and insightful comments from this reviewer. We tried our best to answer these questions.

Comment 1: *Lines 184- 192: It would be great if the authors can comment / clarify the following: In line 31-32 they claimed that cell wall degrading enzymes facilitate pathogen infection. Here they show that simultaneous loss of MoCELL12A and B results in enhanced virulence fig 1 D while MoCELL12A overexpression leads to reduced virulence fig 1e? Then in figure 2C the authors show lesion phenotypes in MoCELL12A over expression lines, which seem to be different lines than the ones used in Figure 1.*

Response: Thanks for this comment. We feel that probably the figure legends of Fig 1d/e and Fig 2c were not clearly stated, and caused little confusing. In Fig 1d/e, the WT rice plants were challenged with *M. oryzae* Δ MoCel12A/B mutant and overexpression strains; while in Fig 2a and 2c, the rice plants are transgenic plants ecotopically expressing MoCel12A, which showed "auto-immune" like phenotype. They are different plants and treatments. We now rephrased the legends to make them clear for readers.

Comment 2: *Figures 3b-c and 4b-c ROS assays: I am surprised that the shape of the ROS bursts induced by cell wall extracts and the specific oligosaccharides look so different. I would assume that they should be more similar even assuming we may be looking at different concentrations of the compounds inducing them etc.*

Response: Thanks for this comment. Indeed, in Fig 4b/c, the oligosaccharides concentrations (10 μ M) are much higher than the cell wall-degraded one (Fig 3b/c), judged by the value of RLU. Although sometimes the ROS responses are little variable, however, we found that lower concentrations of elicitors activated little slower response than higher concentrations, and the signal attenuation was slower in higher concentrations of elicitor-treated rice suspensions. The similar patterns for ROS burst and pH alkalization are often seen triggered by elicitors (Felix et al., 1999, Plant J; Shang-Guan et al., 2018, Plant Physiol)

Minor points:

Comment 3: *The text should be proofread by an English native speaker since in some instances the wording / clarity needs to be improved. Example: line 60, lines 72-74, line 96, line 114, line 134, 163*

Response: We are sorry for this. The revision has been slightly proofread by a native English speaker.

Comment 4: *Line 142: I disagree with the authors. expression of MoCELL12C: expression looks elevated to me from 8-48h compared to mycelium and 72h time point.- for completeness sake the authors should highlight in the discussion section a recent*

publication by Melida et al., 2020

Response: We recognize our original description on MoCel12C was not very precise. When the expression levels (mostly in hyphae) were compared with mycelium but not spores, they were indeed higher. We now rephrased the statement as “By contrast, *MoCel12C* expression was relatively high in spores and infection hyphae at 8 hpi compared to that in mycelium and the later infection stage, suggesting that *MoCel12C* was also involved in infection or pathogen growth”. We thank the reviewer. In addition, the suggested article is about xylan hemicellulose, which should be talked in our study. As suggested, we cited and discussed the article in the discussion.

Reviewer #3:

*This is an interesting work showing that two glycoside hydrolase (GH) proteins expressed by the hemibiotrophic fungal pathogen *Magnaporthe oryzae* of rice can release oligosaccharides from rice cell walls, leading to induction of defense responses. It is a very exhaustive work with a large amount of high quality data, and is beautifully written. Having said that, the MS as a whole avoids putting the proposed mechanism into biological context, and I have some fundamental questions about this. I think the individual parts of the proposed mechanism are well established, from DAMP generation to responses to perception mechanism, but I'd like to make the following points:*

Response: Thanks for the positive and insightful comments. We tried our best to answer this reviewer's questions.

Comment 1: *My main point is; if this mechanism works as proposed, then it is highly disadvantageous for *M. oryzae* to possess *MoCel12A/B*, and there would be strong selection against these genes. Why then are they not rapidly deleted from the *M. oryzae* genome?*

Response: This is a very insightful comment. Indeed, it is little surprising. MoCel12A/B are putative cell wall degrading enzymes (CWDEs) which are supposed to target plant cell wall for degradation. In general, CWDEs help pathogen proliferation by either providing nutrition/helping initial invasion or killing cells for necrotroph pathogens. However, if the enzymes themselves or the damages caused by these enzymes were recognized by hosts, then these CWDEs help in the opposite way. This scenario is often seen for the avirulence function of pathogen effectors. They only help pathogen infection on the susceptible hosts (no recognition). In our case, MoCel12A/B caused damages, and the damages were recognized by OsCERK1, leading to immune response. In fact, pattern recognition receptors (PRRs) vary in different ecotypes and/or genotypes, which could lead to different response to a given PAMP or DAMP. To reinforce the conclusion, in addition to the pretreatment (Fig 4g), we showed that *oscerk1* mutant was susceptible to *M. oryzae*, and pretreatment with the active oligosaccharides did not enhance the resistance in the mutants (Fig S8f). Therefore, we think the susceptible hosts that cannot effectively recognize MoCel12A-caused damages maintain these CWDEs in the genome during evolution.

Comment 2: *The authors proposed that *MoCel12A/B* are required for cell wall degradation during infection. However, the double knockout is apparently able to infect just fine, which suggests to me that there is nothing wrong with its ability to degrade cell walls. I also wonder*

if their biomass assay is appropriate to make the claim of higher virulence, or if they should be counting spore numbers.

Response: Because *M. oryzae* genome carries hundreds of glycan hydrolases (GHs) (<http://www.cazy.org/e3.html>), MoCel12A/B function could be substituted by other GHs, for example by GH family 7. In fact, just few reports showed that knocking out one or two CWDEs could significantly affect pathogen virulence (Doehlemann and Hemetsberger, *New Phytologist*, 2013; Doehlemann et al., *J Plant Physiol*, 2008). However, the immune recognition is dominant in the interaction (MoCel12A/B enzymatic activity related pathogen virulence vs. immune recognition), as overexpression strains (MoCel12A-OE) showed reduced virulence on rice plants (Fig 1e).

For the fungal biomass assays, it is widely accepted and reliable for assessing *M. oryzae* virulence. Unlike the condition for powdery mildew sporulation, the condition for *M. oryzae* sporulation is little challenging and sporulation quantity is not very reproducible. First, it requires over 93% humidity and strict darkness induction. Second, the fungus continuously grows in plant tissue, which affects the sporulation counting. Third, 1~2 h darkness can trigger the release of spores; however, the upright rice leaves make it even harder to collect spores timely. Last, the lesions can reproduce spores up to 20 times. Therefore, the sporulation condition makes the experiments hard to be well controlled and the results are often very variable.

Comment 3: *The authors say somewhat vaguely that this degradation activity occurs during the necrotrophic phase, but the genes are expressed early during infection when we expect the biotrophic phase. Also, DAMP perception would to my mind be active against biotrophs, but would assist necrotrophs.*

Response: We recognize the reviewer's concern. Theoretically, CWDEs are expected to be expressed in the necrotrophic stages. It is true that many CWDEs are expressed at later infection stage (for example, switching to necrotroph stage), which break down host cells for nutrition (Gibson et al., *Curr. Opin. Microbiol.*, 2011). However, MoCel12A/B in our case is likely deployed for acquiring nutrition from plant cell wall at early infection stage. Because these enzymes target hemicellulose and hemicellulose is not the major components of rice cell wall, they probably did not cause host cell death rapidly.

In addition, DAMP perception would trigger plant immune response, especially to defend against biotrophs and hemibiotrophs. However, they may not assist necrotrophs because unlike effector triggered immunity, PTI seldom cause severe cell death (the hypersensitive response). Therefore, PTI typically cannot assist necrotrophs.

Comment 4: *The second point that I wish to make is despite the comprehensive experimentation and data in the MS, the authors have avoided investigating whether MoCel12A/B is active on fungal cell walls, for example as part of cell wall remodelling necessary for growth. I think it is essential to show that they do not degrade fungal cell walls generally, or chitin in particular. Almost all of their data can be explained by a model in which MoCel12A/B release a fungal PAMP that is perceived by CERK1 - which would be highly consistent with current models of chitin (etc) perception by this PRR. (The exception to this is the dwarf and CDR phenotypes of the MoCel12 rice OEX lines, but the expression of the*

transgenes is so high in these lines that confounding effects are certainly possible).

Response: We recognized this concern. Cell wall stress assays imply that MoCel12A/B is not required for the stress response (Fig S3). MoCel12A has been shown to prefer the β -1,3-1,4 glucan as substrate (Takeda et al., 2010), which barely exists in fungal cell walls. Nevertheless, to exclude this possibility as this reviewer suggested, we also provided additional data. Fig S5d showed that the extract of MoCel12A-treated fungal cell wall could not further enhance immune response in rice cells. In addition, MoCel12A/B cannot degrade chitin (Fig S1f).

As this reviewer mentioned, the additional evidence is that the enzymatic activity of MoCel12A is required for MoCel12A-triggered immune response in transgenic plants (Fig 2). This is unlikely due to the high expression levels as WT MoCel12A was expressed at relatively lower levels than that of the mutated MoCel12A in transgenic plants (Fig S5a and S5b). In fact, the high levels of expression of *MoCel12A* and the mutant is due to ectopic expression and the relative to the blank background. Collectively, these data demonstrate that MoCel12A-triggered immune response is unlikely by degrading fungal cell wall.

Specific comments:

Comment 5: *L122 "Similar DAMPs likely also exist in other Poaceae species". I am puzzled by this sentence, as the current MS doesn't seem to shed light on this. I suggest deleting it.*

Response: We recognize the statement is not accurate. We removed the sentence.

Comment 6: *L127 "In the M. 128 oryzae genome, three homologous GH12 endoglucanase genes, namely MoCel12A, 129 MoCel12B, and MoCel12C corresponding to MGG_00677, MGG_08537, and 130 MGG_10972, respectively, were identified". I don't understand here what was done or what the situation is with GH genes in M. oryzae. How many GH families and genes are encoded by M. oryzae? Why were these genes in particular chosen? I can see that MoCel12A and B are closely related, but 12C does not seem to be? What is the amino acid identity between the proteins?*

Response: In our previous study, we sequenced the genomes of two *M. oryzae* strains (Cao et al., 2017, Sci China Life Sci). We found MoCel12A/B/C, which are the only three GH12 family in the genome. Because Takeda et al. (Appl Microbiol Biotechnol, 2010) had cloned the three genes and characterized the enzyme activity of MoCel12A/B, we just cited the reference. We realized the description is little confusing. To make it clear, we now rephrased it as "In the *M. oryzae* genome, three homologous GH12 endoglucanase genes, namely MoCel12A, MoCel12B, and MoCel12C corresponding to MGG_00677, MGG_08537, and MGG_10972, respectively, were identified by Takeda et al. (2010)".

In addition, it was predicted over 260 GHs families in *M. oryzae* genome (<http://www.cazy.org/>). We chose these genes because MoCel12A expressed at higher levels during early infection. MoCel12A/B/C are endoglucanases. They belong to GH family 12. However, MoCel12C is little distant from A and B (Fig S1a). MoCel12A is in GH family 12-1, and MoCel12C is in GH family 12-2 (Takeda et al., 2010). The blastp show that A and B share similarity of 43.60%; A and C share similarity of 32.38%, at protein levels. We included this information in the revision.

Comment 7: *Fig 1/S1 What is the reference gene for RT-qPCR? It is important to know if*

these genes are expressed when the fungi are grown on synthetic media, please add those data (or is this what "mycelium" refers to? - if so, please make it clear).

Response: The reference gene of RT-qPCR is *MoActin* (MGG_03982) of the fungi (in primer list). In addition, the mycelium and spores were grown on the synthetic media. We included the information in the legend and the primer list in the revision.

Comment 8: *Fig 1C and 1D need complementation lines.*

Response: The data of complementation strains were included, where the virulence in Fig 1c and 1d was suppressed (Fig S4).

Comment 9: *L132 "...because CWDEs function to break down host cell walls, facilitating necrotrophic growth" please reference this statement. I don't understand it; necrotrophic growth occurs after the fungus has penetrated the cell wall, please comment? It is a very important point for the overall model (see my general comments).*

Response: Conceptually, CWDEs help pathogen to break down plant cell walls for necrotrophic pathogens (Kubicek et al., Annu.Rev.Phytopathol.2014 ; Gibson et al., Curr. Opin. Microbiol., 2011). Indeed, the penetration typically occurs at 4~6 hpi, but the expression of MoCel12A/B peaked at 24 hpi/12 hpi (Fig 1A), suggesting that MoCel12A/B is likely deployed for acquiring nutrition or starting to break down host cells after initial infection. In either cases, it helps to break down the cells and potentially facilitate the transition to necrotroph, although the cell death can often be seen at 24 hpi. In fact, it is hard to tell when the necrotroph stage starts, but surely occurs after the initial invasion. However, we agree with the reviewer and recognize the original description is not precise. To exclude the misunderstanding, we now rephrased the statement as "because CWDEs function to break down host cell walls, they are considered to proceed necrotrophic growth generally; however, these genes were induced during early infection (by 24 hours post inoculation, hpi), a stage of biotrophic growth."

For the roles of CWDEs, the biotroph *Blumeria graminis* and *Ustilago maydis*, for example, lack many CWDEs but they are abundant in hemibiotroph *M. oryzae* and necrotroph *Fusarium graminearum* and *B. cinerea* etc. (reviewed by Gibson et al., Curr. Opin. Microbiol., 2011, and Kubicek et al., Annu. Rev. Phytopathol, 2014). Many other CWDEs are expressed at later infection stage (Gibson et al., Curr. Opin. Microbiol., 2011), which may collectively result in the cell death and the transition of biotroph to necrotroph in *M. oryzae*. We have cited these references in the revision.

Comment 10: *Fig 2B Why were only two potential substrates used for these tests? I would like to see many more, particularly fungal polymers such as chitin or chitosan, and also fungal cell wall preparations. That is because the alternative hypothesis is that MoCel12A/B release a fungal PAMP. Secondly, how do the activities in Fig 2B compare to those of known B-glucanases?*

Response: The reason that using β -glucan and xyloglucan as substrates is because MoCel12A/B/C are predicted as β -endoglucanases in GH family 12, and the substrate preference has been reported by Takeda et al. (2010). As we respond to comment 4, chitin is not the substrate of MoCel12A (Fig S1f), and MoCel12A-digested *M. oryzae* cell wall cannot

trigger immune response in rice cells (Fig S5d), excluding the possibility that the immune activation is due to *M. oryzae*-derived PAMPs in this study. In addition, we also tested several other substrates, the carboxymethyl-cellulose (CMC) and crystalline cellulose, which are not the substrates of MoCel12A/B either (Fig S1e).

“Fig2B” should be “Fig1B”? In general, the β -glucanases could be classified to specifically hydrolyze β -1,4-glucan, β -1,3-glucan, or β -1,3-1,4-glucans, but the GH family 12 prefer β -1,4-glucans or β -1,3-1,4-glucan (Okada et al., Appl Environ Microbiol, 1998; Takeda et al., 2010). In addition to Fig 1B, we further confirmed that MoCel12A specifically hydrolyzed β -1,3-1,4-glucans but not others (Fig S1e and S1f) in the revision.

Comment 11: *L160 Would be good to mention here that the rice plants expressing Magnaporthae Mol12A are stunted.*

Response: We thank this comment. L160 states the enzyme activity and the key residues for the activity. We mentioned the stunted phenotype in L221 (Fig 2a).

Comment 12: *L163 “MoCel12A/B are hydrolases that target plant cell walls for degradation” I find this statement a bit strong for this stage of the MS, please tone down (also, the double knockout seems to degrade cell walls just fine judging from Fig 1D).*

Response: We agree with this comment. It was rephrased as “MoCel12A/B are putative CWDEs that are assumed to target plant cell wall for degradation”. *M. oryzae* carries over 260 GHs (predicted by <http://www.cazy.org/>), many of them are β -1,4-, β -1,3- or β -1,3-1,4-glucanases. Thus, in the double knockout, MoCel12A/B function may be substituted by other glucanases and should not significantly impair the virulence, theoretically. As responding to comment 2, just few reports showed that knocking out 1 or 2 CWDEs could significantly impair pathogen virulence (Doehlemann and Hemetsberger, New Phytologist, 2013; Doehlemann et al., J Plant Physiol, 2008; Gibson et al., Curr. Opin. Microbiol., 2011).

Comment 13: *L162/Fig S2B what do the Congo Red and SDS assays tell us? The MoCel12A/B double mutant should have been tested using these assays.*

Response: The Congo Red and SDS are often used as fungal cell wall stress reagents, which interfere with the construction and stress response of the cell wall (Ram and Klis, Nat. Protoc., 2006; Igual et al., EMBO J, 1996). For example, Congo Red preferentially stains chitin in the cell wall of fungi, thus it is thought to interfere with cell wall assembly by binding to chitin. As a result, the cell wall-weakening effect by Congo red activates the cell wall stress response, including transcriptional activation of many genes encoding proteins that have cell wall-reinforcing functions (Ram and Klis, Nat. Protoc., 2006). Therefore, these reagents can be used to examine the cell wall stress response and to examine if there is a cell wall integrity defect in the MoCel12A/B mutants. In the revision, MoCel12A/B double mutant has been included in the cell wall stress assays (Fig S3d).

Comment 14: *L188 “enhanced virulence”...I think you have to be careful with these terms. I accept that there appears to be greater fungal biomass (although I'd like to see a chitin assay too), but for greater virulence I'd like to see spore counts - this should be possible?*

Response: As we respond to comment 4, chitin is not the substrate of MoCel12A/B. Also in the response to comment 2, we listed the reasons why spore counts are not appropriate for this assay. However, we agree with the comment. The more appropriate description might be “enhanced pathogenicity” as there were more fungal growth and lesions in rice tissue.

Comment 15: *L221 “MoCel12A-activated rice immune responses could be associated with MoCel12A as a PAMP or with MoCel12A-hydrolyzed cellooligomers from rice cell walls as DAMPs.” The third hypothesis is that MoCel12A releases a fungal PAMP.*

Response: We did not include the third hypothesis in original submission is because MoCel12A/B unlikely target fungal cell wall, as β -1,3-1,4- glucan mainly exists in grass plants. However, since we have provided additional data using fungal cell wall and chitin as substrate for the enzymes (Fig S1f and S5d), to make it clear and easier to be followed for readers, we therefore include the third hypothesis in the revision.

Comment 16: *If you can knock MoCel12A/B out and enhance virulence, what stops them being lost?*

Response: As we respond earlier to the comment 1 and others, we believe all the “disadvantage” genes function in those hosts that cannot effectively “recognize” them. In other words, these genes are helpful for infection only on the susceptible hosts. For example, the fungi are more virulent on *cerk1* mutant (Fig S8f). Just like pathogen effectors, we believe these “disadvantage” genes are maintained by susceptible hosts in nature.

Comment 17: *Fig 4G/H- 100 uM BGTRIB/BGTETC seems very high to me, please comment? Any way to relate this to in vivo conditions under M. oryzae infection? (kind of discussed in L298-301, but concentrations are not mentioned: can the data in Fig S5C/D be converted into comparable concentrations?).*

Response: The revised Fig S6b and S6c showed that 10uM BGTRIB generated a typical ROS burst. To make sure a better treatment, we typically use 10 folds higher of the concentrations for the elicitors on rice leaves than that used in ROS burst assays (suspension cells), because rice leaf surface is very hard. In fact, for the elicitor treatment on plant leaves, the concentration is often higher (500ug/ml chitin, Faulkner et al., PNAS, 2013). Importantly, 100 uM BGTRIB/BGTETC did not cause side effects on WT plants (Fig S8f). In fact, in our hand, 10 uM BGTRIB/BGTETC suppressed the fungal growth by the average reduction of 30% (data not shown), but the data are variable, possibly due to the hard surface of the rice leaves.

In the revised Fig S7, the apoplastic BGTRIB can be converted to ~5uM for the OE strain-infected, and ~2 uM for the Guy11-infected in 1g rice leaves. The concentration was estimated by extracting the apoplastic fluids. It is low compared to the exogenous treatment though. However, within 48 hs, the fungi did not spread to all the apoplastic space. The exact infection sites should have higher concentration.

Comment 18: *Discussion would be improved by more of a biological discussion of how this particular mechanism interfaces with rice infection by M. oryzae.*

Response: We tried our best to improve the discussion as suggested in the revision.

REVIEWER COMMENTS

Reviewer #1 (Remarks to the Author):

The revised version of manuscript by Yang et al. has been significantly improved.

The reviewer 1 understood that cellobiose is not released in the hydrolysis by MoCel12A and does not cause ROS generation at least in rice suspension cultured cells. With regard to the dimerization of receptors, OsCERK1 and OsCeBiP, by cell wall oligosaccharides other than b-1,3:1,4-glucan oligosaccharides, the reviewer 1 understood that xylooligosaccharide also induces dimerization but its affinity to ECD was lower than b-1,3:1,4-glucan oligosaccharides. I do not require to include this additional result on other oligosaccharides in the manuscript and agree that the result does not affect main statements such as that b-1,4-1,3-glucose-based oligosaccharides are the potential ligands of OsCERK1 and open a new avenue for OsCERK1 in immune activation in lines 497-499.

The words, b-1,4-1,3-glucose and b-1,4-1,3-glucose, may be confused. All b-1,3:1,4-glucan oligosaccharides should be abbreviated with easier ones such as G4G3G and G3G4G and mentioned with them throughout the text. In particular, 'b-1,3-1,4-glucose (β -Glc-1,3- β -Glc-1,4-Glc)-based oligosaccharides' in the Abstract is difficult to understand. Is it G3G4G oligosaccharide or oligosaccharide including b-1,3- and b-1,4-linked glucosyl residues?

Reviewer #2 (Remarks to the Author):

Th authors have addressed my concerns in an adequate manner.

Reviewer #3 (Remarks to the Author):

This revised MS is improved with some additional data and writing. However I am very disappointed that the authors have not attempted to put their results into biological context, as criticized in my comments 1-3 of my previous review. The substantive points are these:

A. It makes no sense for *M. oryzae* to express MoCel12A/B, because they induce immunity via CERK1 and are dispensable for infection. This is particularly relevant in view of the fact that there are probably *M. oryzae* effectors that act redundantly with MoCel12A/B, as pointed out by the authors in their response to my previous comment 2.

B. These effectors are expressed too early in infection to make sense with their model of infection. This fact reinforces point A.

These two points should be described and discussed in the Discussion section.

I have made a list of suggested writing changes below (not comprehensive) - I hope the authors find these helpful.

L44 change to "that was largely compromised"

L47 change to "rice cell wall"

L47 change to "by an OsCERK1 and"

L60 delete "where"

L63 peptidoglycan is misspelt.

L72 "as", not "at"

L110 change to "walls", "responses"

L158 change "with" to "fused to a", same for L164
L167 change "as" to "that of"
L168 change "and" to "or"
L180 "target the plant cell wall"
L181 change "must" to "should"
L203 delete "response" (or change to "responses")
L210 "more severe", not "severer"
L248 "fungal cell walls", not wall.
L255 "cell walls" not wall
L255 "an immune response" or "immune responses"
L257 "walls"
L293 "responses".
L302 "beginning with a"
L321 "responses"
L330 delete "presence"
L332 "fold" not "folds". Also L334
L333 "in" not "of"
L342 "the ROS"

349 "bursts were"
L367 "this" not "the"
L368 "the oscrk1", also L369 and L370 and L375
L374 "the ROS"
L377 "to" not "as"
L379 "of" not "on"
L416 "bursts"
L417 what does "resembling" mean in this sentence? Overall the sentence should be rewritten, it is very complex with too many commas.

Reviewer #1 (Remarks to the Author):

Comment 1: The revised version of manuscript by Yang et al. has been significantly improved.

The reviewer 1 understood that cellobiose is not released in the hydrolysis by MoCel12A and does not cause ROS generation at least in rice suspension cultured cells. With regard to the dimerization of receptors, OsCERK1 and OsCeBiP, by cell wall oligosaccharides other than b-1,3:1,4-glucan oligosaccharides, the reviewer 1 understood that xylooligosaccharide also induces dimerization but its affinity to ECD was lower than b-1,3:1,4-glucan oligosaccharides. I do not require to include this additional result on other oligosaccharides in the manuscript and agree that the result does not affect main statements such as that b-1,4-1,3-glucose-based oligosaccharides are the potential ligands of OsCERK1 and open a new avenue for OsCERK1 in immune activation in lines 497-499.

The words, b-1,4-1,3-glucose and b-1,4-1,3-glucose, may be confused. All b-1,3:1,4-glucan oligosaccharides should be abbreviated with easier ones such as G4G3G and G3G4G and mentioned with them throughout the text. In particular, 'b-1,3-1,4-glucose (β -Glc-1,3- β -Glc-1,4-Glc)-based oligosaccharides' in the Abstract is difficult to understand. Is it G3G4G oligosaccharide or oligosaccharide including b-1,3- and b-1,4-linked glucosyl residues?

Response: We thank this review's support, and we have replaced the b-1,4-1,3-glucose etc. with G4G4G as suggested.

Reviewer #2 (Remarks to the Author):

The authors have addressed my concerns in an adequate manner.

Response: We thank this review's support.

Reviewer #3 (Remarks to the Author):

Comment 1: This revised MS is improved with some additional data and writing. However I am very disappointed that the authors have not attempted to put their results into biological context, as criticized in my comments 1-3 of my previous review. The substantive points are these: **A.** It makes no sense for *M. oryzae* to express MoCel12A/B, because they induce immunity via CERK1 and are dispensable for infection. This is particularly relevant in view of the fact that there are probably *M. oryzae* effectors that act redundantly with MoCel12A/B, as pointed out by the authors in their response to my previous comment 2. **B.** These effectors are expressed too early in infection to make sense with their model of infection. This fact reinforces point A. These two points should be described and discussed in the Discussion section.

Response: We thank this suggestion and we are very sorry for not fully understanding this reviewer's previous comments. The main issue by our understandings is that why *M. oryzae* expresses MoCel12A/B since they activate the immunity. We now realize that it is a very important point of our study, and discussion cannot fully address this issue. In fact, MoCel12A/B weakly contribute to pathogenicity. In addition to the previous data (Fig S8F),

we now provide additional data showing that overexpression of *MoCel12A* in *M. oryzae* could slightly but significantly facilitate infection in the *oscerk1* mutant but reduce infection in WT plants (Fig S8G). These data indicate that MoCel12A/B contribute to pathogenicity if there is no immune activation.

In addition to these new data, we also discuss why these “disadvantage” genes “survived” during evolution. Except the host specificity (for example, CERK1 polymorphism in different rice varieties), these “damps”- oligosaccharides may be modified by either host or *M. oryzae* oxidases. This case is similar to the chitin deacetylation, which can shield chitin from recognition by CERK1 in plants (Gao et al., 2019, Nat Plants). Flagellin could be deglycosylated to evade host recognition as well (Buscaill et al, 2019, Science). There are evidences showing that similar oligosaccharides could be oxidized and would be inactive after oxidation (Locci et al., 2019, Plant J), although it has not been proved in our case yet. However, it is worth future investigating because both *M. oryzae* and rice genomes carry the homolog genes.

For the question B, Fig 1A and 1B showed that their highest expression levels at 12~24 hpi. Therefore, they could act for acquiring nutrition or simply degrading the cell wall. Either way would release the DAMPs and lead to the immune activation in rice. Because there is no clear border to divide biotroph stage from necrotroph stage for this hemibiotrophic pathogen, early expression of the CWDEs may cause host cell death gradually. In other word, it is a time course. However, whatever for nutrition purpose or destroying host cells, they could be explained by our model. We have discussed these possibilities in the discussion.

Editorial comments: I have made a list of suggested writing changes below (not comprehensive) - I hope the authors find these helpful.

L44 change to “that was largely compromised”

L47 change to “rice cell wall”

L47 change to “by an OsCERK1 and”

L60 delete “where”

L63 peptidoglycan is misspelt.

L72 “as”, not “at”

L110 change to “walls”, “responses”

L158 change “with” to “fused to a”, same for L164

L167 change “as” to “that of”

L168 change “and” to “or”

L180 “target the plant cell wall”

L181 change “must” to “should”

L203 delete “response” (or change to “responses”)

L210 “more severe”, not “severer”

L248 “fungal cell walls”, not wall.

L255 “cell walls” not wall

L255 “an immune response” or “immune responses”

L257 “walls”

L293 “responses”.

L302 “beginning with a”

L321 “responses”

L330 delete “presence”

L332 “fold” not “folds”. Also L334

L333 “in” not “of”

L342 “the ROS”

L349 “bursts were”

L367 “this” not “the”

L368 “the oscar1”, also L369 and L370 and L375

L374 “the ROS”

L377 “to” not “as”

L379 “of” not “on”

L416 “bursts”

L417 what does “resembling” mean in this sentence? Overall the sentence should be rewritten, it is very complex with too many commas.

Response: We very much appreciate this reviewer’s comments on the writing, which greatly improve the quality of our manuscript.

REVIEWERS' COMMENTS

Reviewer #3 (Remarks to the Author):

Thank you for patiently addressing my concerns. I am now satisfied that the MS is ready for publication.